# Earlier Alzheimer's disease onset is associated with tau pathology in brain hub regions and facilitated tau spreading

Lukas Frontzkowski[1], Michael Ewers [1,2], Matthias Brendel [3,4], Davina Biel[1], Rik Ossenkoppele [5,6], Paul Hager [1], Anna Steward[1], Anna Dewenter [1], Sebastian Römer [1,7], Anna Rubinski[1], Katharina Buerger[1,2], Daniel Janowitz[1], Alexa Pichet Binette [5], Ruben Smith [5,8], Olof Strandberg[5], Niklas Mattsson Carlgren[5,8], Martin Dichgans [1,2,4], Oskar Hansson [5,9] & Nicolai Franzmeier [1,4] ✉

In Alzheimer's disease (AD), younger symptom onset is associated with accelerated disease progression and tau spreading, yet the mechanisms underlying faster disease manifestation are unknown. To address this, we combined resting-state fMRI and longitudinal tau-PET in two independent samples of controls and biomarker-confirmed AD patients (ADNI/BioFINDER, $n = 240/57$). Consistent across both samples, we found that younger symptomatic AD patients showed stronger tau-PET in globally connected fronto-parietal hubs, i.e., regions that are critical for maintaining cognition in AD. Stronger tau-PET in hubs predicted faster subsequent tau accumulation, suggesting that tau in globally connected regions facilitates connectivity-mediated tau spreading. Further, stronger tau-PET in hubs mediated the association between younger age and faster tau accumulation in symptomatic AD patients, which predicted faster cognitive decline. These independently validated findings suggest that younger AD symptom onset is associated with stronger tau pathology in brain hubs, and accelerated tau spreading throughout connected brain regions and cognitive decline.

Sporadic Alzheimer's disease (AD) is highly heterogeneous, with variable clinical expression, age of symptom onset as well as cognitive and neuropathological trajectories[1,2]. In sporadic AD, younger age of symptom onset has been previously associated with accelerated tau accumulation[3,4] and neurodegeneration[5], faster cognitive decline[6–8] and higher mortality[9], together suggesting that younger symptom onset is potentially driven by a more aggressive form of AD[9,10]. Yet, it is

unclear why sporadic AD patients with younger symptom onset show faster pathological[3–5] and clinical progression[6–8] than patients with later symptom onset. From a histopathological point of view, non-mendelian early- and late-onset sporadic AD share the same molecular pathologies including amyloid-beta (Aβ) plaques and neurofibrillary tangles composed of hyperphosphorylated 3/4R tau pathology, suggesting that an earlier symptom onset is unlikely to be specifically

---

[1]Institute for Stroke and Dementia Research (ISD), University Hospital, LMU Munich, Munich, Germany. [2]German Center for Neurodegenerative Diseases (DZNE), Munich, Germany. [3]Department of Nuclear Medicine, University Hospital, LMU Munich, Munich, Germany. [4]Munich Cluster for Systems Neurology (SyNergy), Munich, Germany. [5]Clinical Memory Research Unit, Department of Clinical Sciences Malmö, Lund University, Lund, Sweden. [6]Alzheimer Center Amsterdam, Department of Neurology, Amsterdam Neuroscience, Vrije Universiteit Amsterdam, Amsterdam UMC, Amsterdam, The Netherlands. [7]Department of Neurology, University Hospital, LMU Munich, Munich, Germany. [8]Department of Neurology, Skåne University Hospital, Lund, Sweden. [9]Memory Clinic, Skåne University Hospital, Lund, Sweden. ✉e-mail: Nicolai.franzmeier@med.uni-muenchen.de

driven by distinct molecular characteristics of AD pathology[11]. Rather, previous post-mortem and tau-PET imaging studies reported that an earlier AD symptom onset is associated with a different spatial distribution pattern of tau pathology[10,12], i.e., the key driver of neurodegeneration[13] and cognitive decline in AD[14,15]. Specifically, post-mortem assessments in AD patients found that a more pronounced neocortical and hippocampal-sparing pattern of neurofibrillary tau tangle pathology is associated with younger age at symptom onset and faster ante-mortem cognitive decline, whereas a spatially more restricted limbic-predominant pattern of neurofibrillary tau tangles was associated with older age at symptom onset and slower ante-mortem cognitive decline[10]. Similarly, tau-PET studies in AD patients found that younger age in general[12,16] and early-onset AD (i.e., before the age of 65) in particular are associated with stronger tau-PET uptake in fronto-parietal association cortex regions[12,17–19] with relative sparing of medial temporal lobe regions[18]. Together, these findings suggest that an earlier symptom onset in AD is associated with a pattern shift of tau pathology deposition from allocortical medial temporal lobe regions toward neocortical fronto-parietal association cortices.

As shown by functional MRI studies, fronto-parietal brain regions harbor key hubs that are globally connected to the rest of the brain[20,21] and central for cognitive function[22,23] as they facilitate information integration across different brain networks during cognitive demands[22]. In a series of resting-state fMRI studies, we and others could previously show that the functional integrity of fronto-parietal control network hubs is critically important for maintaining cognitive function in aging[24], AD[25–27] and other neurodegenerative diseases[28]. Since tau pathology has been shown to impair neuronal function[29] and to drive neurodegeneration[13], a stronger tau pathology load in fronto-parietal hub regions that are highly relevant for cognition may thus drive earlier symptom manifestation in AD. In addition, tau pathology deposition in globally connected hub regions may further accelerate the progression and spreading of tau pathology itself, which is in turn a strong driver of cognitive decline[30]. Specifically, preclinical studies have consistently shown that tau spreads trans-synaptically across interconnected neurons[31,32]. Similarly, we and others reported previously in combined tau-PET and MRI studies that tau spreads preferentially across functionally and anatomically connected brain regions, where the connectivity pattern of tau harboring epicenter regions determines the subsequent spreading pattern of tau pathology[2,33–38]. Thus, the occurrence of tau pathology in globally connected hubs early in the course of AD may ensue faster and more widespread tau spreading thereby driving earlier disease manifestation and faster clinical progression[14,15,30].

To address these open questions, the major aims of the current combined tau-PET and resting-state fMRI study were to assess whether: (1) younger age is associated with stronger tau pathology in globally connected hub regions compared to non-hubs in patients with symptomatic AD; (2) whether tau pathology in hub regions is associated with accelerated subsequent tau spreading and; (3) whether the association between younger age and faster tau accumulation rates in symptomatic AD[3] is mediated by stronger tau pathology in globally connected hub regions. To this end, we employed two independent samples covering the entire AD spectrum, including 240 participants of the Alzheimer's disease neuroimaging initiative (ADNI) and 57 subjects of the BioFINDER study for replication, all with available baseline amyloid-PET and longitudinal Flortaucipir tau-PET. To map the topology of globally connected hubs across the brain, we used high-resolution resting-state fMRI data from 1000 healthy participants of the human connectome project (HCP). Based on these resting-state fMRI data, we estimated an atlas-based[39] whole-brain map of the graph-theoretical metric weighted degree (i.e., also referred to as global connectivity)[25] in healthy individuals that are unaffected by AD pathology. By mapping individual tau-PET patterns in AD patients to the topology of globally connected brain hubs, we determined the

degree to which individual tau-PET patterns are expressed in globally connected hub regions in the fronto-parietal association cortex vs. weakly connected non-hub regions e.g., in temporo-limbic and visual cortex region. Based on these data, we show that (1) individual tau-PET deposition patterns are indeed stronger in globally connected hub regions in younger patients with symptomatic AD and associated with earlier symptom onset; (2) that a more hub-like pattern of tau pathology deposition at baseline is associated with accelerated subsequent annual tau accumulation and; (3) that the association between younger age and faster tau accumulation rates in symptomatic AD is mediated by a more hub-like tau-PET pattern thereby driving faster cognitive decline.

## Results
To address and validate the aims of the current study, we included two fully independent samples (ADNI and BioFINDER) with baseline amyloid-PET and longitudinal $^{18}$F-Flortaucipir tau-PET. We further included longitudinal cognitive data for a memory composite as well as age of symptom onset estimates in symptomatic AD patients which were available for a subset in ADNI. Both samples included cognitively normal (CN) control subject without evidence of AD pathology (ADNI/BioFINDER, $n = 93/16$), CN amyloid-positive individuals (i.e., preclinical AD, ADNI/BioFINDER, $n = 60/16$) and patients with symptomatic AD (i.e., mild cognitively impaired (MCI) and AD dementia; ADNI/BioFINDER, $n = 89/25$). Subject demographics for both samples are displayed in Table 1. Resting-state fMRI data from 1000 HCP participants was used to estimate a healthy connectivity template for the Schaefer 200 ROI atlas that covers the neocortex (Fig. 1A)[39]. Across the 1000 HCP participants, we created a group-average functional connectivity matrix, which was density thresholded at 30% to eliminate potentially weak and noisy connections and transformed to connectivity-based distance (i.e., higher functional connectivity = shorter connectivity-based distance, Fig. 1B). For each ROI, we then determined the mean connectivity-based distance to the remaining ROIs as a measure of hub-ness (i.e., shorter distance to the rest of the brain = higher hub-ness, Fig. 1C), which is equivalent to the graph theoretical measure weighted degree. In line with previous studies[20,40], hub regions were primarily found in the fronto-parietal association cortex. To determine a hub map that scales brain regions between hubs and non-hubs, the group-average global connectivity map (Fig. 1C) was rescaled between 1 (i.e., hub) and −1 (i.e., non-hub) (Fig. 1D), in order to assess in a later step whether individual tau-PET patterns matched a hub or non-hub pattern. Tau-PET SUVR data (i.e., intensity normalized to the inferior cerebellar gray) was parcellated using the same 200 ROI parcellation (Fig. 1A), and pre-established two-component Gaussian mixture modeling (Fig. 1E) was applied to transform tau-PET SUVRs to tau-PET positivities in order to separate off-target from on-target tau-PET binding[33]. In an additional exploratory step, we also included tau-PET SUVRs from pre-established hippocampal ROIs[41], which exclude the choroid plexus and therefore minimize the influence of Flortaucipir off-target binding. As for the remaining cortex, hippocampal tau-PET SUVRs underwent two-component Gaussian mixture modeling to further reduce the effect of off-target binding. Surface renderings of baseline tau-PET positivity across diagnostic groups are shown in Fig. 1F for ADNI and for BioFINDER, showing no evidence for tau-PET positivity in amyloid-negative controls, minimally elevated temporal-lobe tau levels in preclinical AD, vs. gradually increasing tau-PET positivity across symptomatic AD patients with MCI and AD dementia.

### Younger age is associated with a more hub-like tau-PET pattern in symptomatic AD
To assess the degree to which individual tau-PET deposition represented a hub or non-hub pattern, we mapped patient-specific tau-PET positivity to the scaled hub map shown in Fig. 1D. Specifically, we multiplied ROI-specific subject-specific tau-positivity with the scaled

**Table 1 | Subject demographics**

| ADNI (n = 240) | CN Aβ− | CN Aβ+ | MCI Aβ+ | AD dementia | p value |
|---|---|---|---|---|---|
| | (n = 93) | (n = 60) | (n = 56) | (n = 31) | |
| Age (M/SD) | 72.54 (6.84) | 75.23 (6.65) | 75.14 (7.63) | 76.87 (7.74) | 0.011 |
| Sex (m/f) | 39/54 | 26/34 | 29/27 | 20/11 | 0.134 |
| Education (M/SD) | 16.51 (2.48) | 16.6 (2.24) | 16.25 (2.56) | 15.9 (2.68) | 0.568 |
| MMSE (M/SD) | 29.21 (0.91) | 28.78 (1.78) | 27.33 (1.92) | 21.75 (5.44) | <0.001 |
| ADNI-MEM (M/SD) | 1.11 (0.59) | 0.87 (0.53) | 0.2 (0.63) | −0.77 (0.66) | <0.001 |
| Centiloid (M/SD) | −3.62 (12.88) | 69.62 (32.63) | 76.72 (29.98) | 89.28 (36.33) | <0.001 |
| ApoE4 (pos/neg/NA) | 29/50/14 | 33/21/6 | 32/10/14 | 12/6/13 | <0.001 |
| Global tau-PET SUVR | 1.08 (0.08) | 1.10 (0.09) | 1.21 (0.22) | 1.38 (0.41) | <0.001 |
| Temporal meta ROI tau-PET SUVR | 1.12 (0.1) | 1.15 (0.1) | 1.29 (0.23) | 1.43 (0.36) | <0.001 |
| Tau hub gradient (M/SD) | −0.12 (0.16) | −0.17 (0.17) | −0.16 (0.19) | −0.08 (0.24) | 0.090 |
| Mean tau-PET follow-up time in years (M/SD) | 2.14 (1.12) | 1.86 (0.71) | 1.70 (0.83) | 1.62 (0.61) | <0.001 |
| BioFINDER (n = 57) | CN Aβ− | CN Aβ+ | MCI Aβ+ | AD dementia | p value |
| | (n = 16) | (n = 16) | (n = 7) | (n = 18) | |
| Age | 73.88 (5.32) | 75.44 (6.09) | 72.71 (6.63) | 69.83 (10.48) | 0.192 |
| Sex (m/f) | 10/6 | 6/10 | 2/5 | 11/7 | 0.245 |
| Education (M/SD) | 12.59 (4.06) | 10.56 (3.22) | 11.14 (2.67) | 13.44 (3.26) | 0.097 |
| MMSE (M/SD) | 29 (1.1) | 29.31 (1.08) | 25.57 (2.94) | 22.06 (5.17)[b] | <0.001 |
| Global Flutemetamol SUVR | 0.52 (0.03) | 0.77 (0.12) | 0.84 (0.14) | 0.97 (0.15) | <0.001 |
| ApoE4 (pos/neg/NA) | 0/16/0 | 10/6/0 | 4/3/0 | 10/7/1 | <0.001 |
| Global tau-PET SUVR | 1.04 (0.05) | 1.05 (0.05) | 1.37 (0.37) | 1.49 (0.36) | <0.001 |
| Temporal meta ROI tau-PET SUVR | 1.08 (0.06) | 1.09 (0.05) | 1.52 (0.39) | 1.56 (0.31) | <0.001 |
| Tau hub gradient (M/SD) | −0.16 (0.22) | −0.09 (0.23) | −0.15 (0.23) | −0.06 (0.20) | 0.566 |
| Mean tau-PET follow-up time in years (M/SD) | 2.03 (0.47) | 1.91 (0.32) | 1.82 (0.12) | 1.97 (0.34) | 0.484 |

p values were derived from ANOVAs for continuous measure sand from Chi-squared tests for categorical measures.
M mean, SD standard deviation, m male, f female, MMSE mini-mental state exam, SUVR standardized uptake value ratio, NA not available.

hub map shown in Fig. 1D, in order to determine hub-weighted tau positivity values, which were subsequently averaged and divided by the overall tau-PET positivity averaged across all 200 ROIs, to adjust for global tau deposition. We specifically adjusted for global tau levels, so that the tau hub ratio specifically reflected the pattern of tau deposition and not the overall tau severity. The mapping of subject-specific tau-PET patterns to the scaled hub map is illustrated in Fig. 1G–M, yielding a numeric index (i.e., henceforth referred to as tau hub ratio) that indicates whether a subject-specific tau-PET pattern is more representative of hub regions (i.e., more positive) or non-hub regions (i.e., more negative), while adjusting for global tau-PET positivity. Using an ANOVA, no difference in the tau hub ratio was found between diagnostic groups (Table 1, ADNI, $p = 0.090$, BioFINDER, $p = 0.566$), highlighting that the tau hub ratio does capture spatial patterns of tau deposition but not increasing diagnostic severity. We then tested whether younger age was associated with a higher tau hub ratio in patients with symptomatic AD (i.e., MCI and dementia), using linear regression controlling for sex, education and diagnosis. Supporting this, we found younger age to be associated with a more positive tau hub ratio in symptomatic AD patients of both samples (ADNI: $\beta = -0.238$, $p = 0.024$, Fig. 2A, right panel; BioFINDER: $\beta = -0.482$, $p = 0.018$, Fig. 2B, right panel). Exact tests using 1000 beta values from null-model tau hub ratios that were generated using shuffled connectomes as a reference confirmed these results (ADNI: $p = 0.016$; BioFINDER: $p = 0.019$). Further, in a subset of 38 symptomatic ADNI participants with available resting-state fMRI data, this association was consistent when using subject-level connectivity data and tau-PET to determine the tau hub ratio ($b = -0.307$, $p = 0.037$). Results also remained significant when repeating the analyses using robust regression or when additionally controlling for global Aβ levels or ApoE4 status (only available in a subset, see Table 1 and see

Supplementary Table 1). In contrast, no association between age and the tau hub ratio was detected for asymptomatic patients with pre-clinical AD (ADNI: $\beta = 0.087$, $p = 0.542$, Fig. 2A, left panel; BioFINDER: $\beta = -0.307$, $p = 0.901$, Fig. 2B, left panel), in which abnormal tau-PET signal was minimal. When repeating the above-described analysis with the actual age of cognitive symptom onset based on informant assessments (i.e., available in 77/89 symptomatic ADNI participants), we found congruent results, showing that younger age of symptom onset was associated with a higher tau hub ratio ($\beta = -0.256$, $p = 0.024$, Fig. 3A), using linear regression controlling for the time difference between age of onset and the tau-PET acquisition date, sex, education and diagnosis. In addition, we tested whether the epicenters of tau pathology (i.e., 10% of brain regions with highest baseline tau-PET positivity,) were more likely to be located in more globally connected regions in younger symptomatic AD patients (see Supplementary Fig. 1 for a mapping of epicenters; anatomical locations of the epicenters are labeled in the source data file). Supporting this, we found that younger age was indeed associated with stronger hub-ness (i.e., shorter connectivity-based distance to the rest of the brain) of tau epicenters in patients with symptomatic AD (ADNI: $\beta = 0.300$, $p = 0 < 0.001$, Fig. 2C, right panel; BioFINDER: $\beta = 0.646$, $p < 0.001$, Fig. 2D, right panel), using linear regression controlling for sex, education and diagnosis. Again, this result pattern was not observed in preclinical AD (ADNI: $\beta = -0.028$, $p = 0.785$, Fig. 2C, left panel; BioFINDER: $\beta = 0.022$, $p = 0.943$, Fig. 2D, left panel), controlling for sex and education. In ADNI, we repeated these analyses with informant-based ages of cognitive symptom onset. Here, we found congruent results, showing that younger age of onset was associated with stronger hub-ness of tau epicenters (i.e., shorter connectivity-based distance to the rest of the brain), controlling for the time difference between age of onset and the tau-PET acquisition date, sex, education and diagnosis ($\beta = 0.352$,

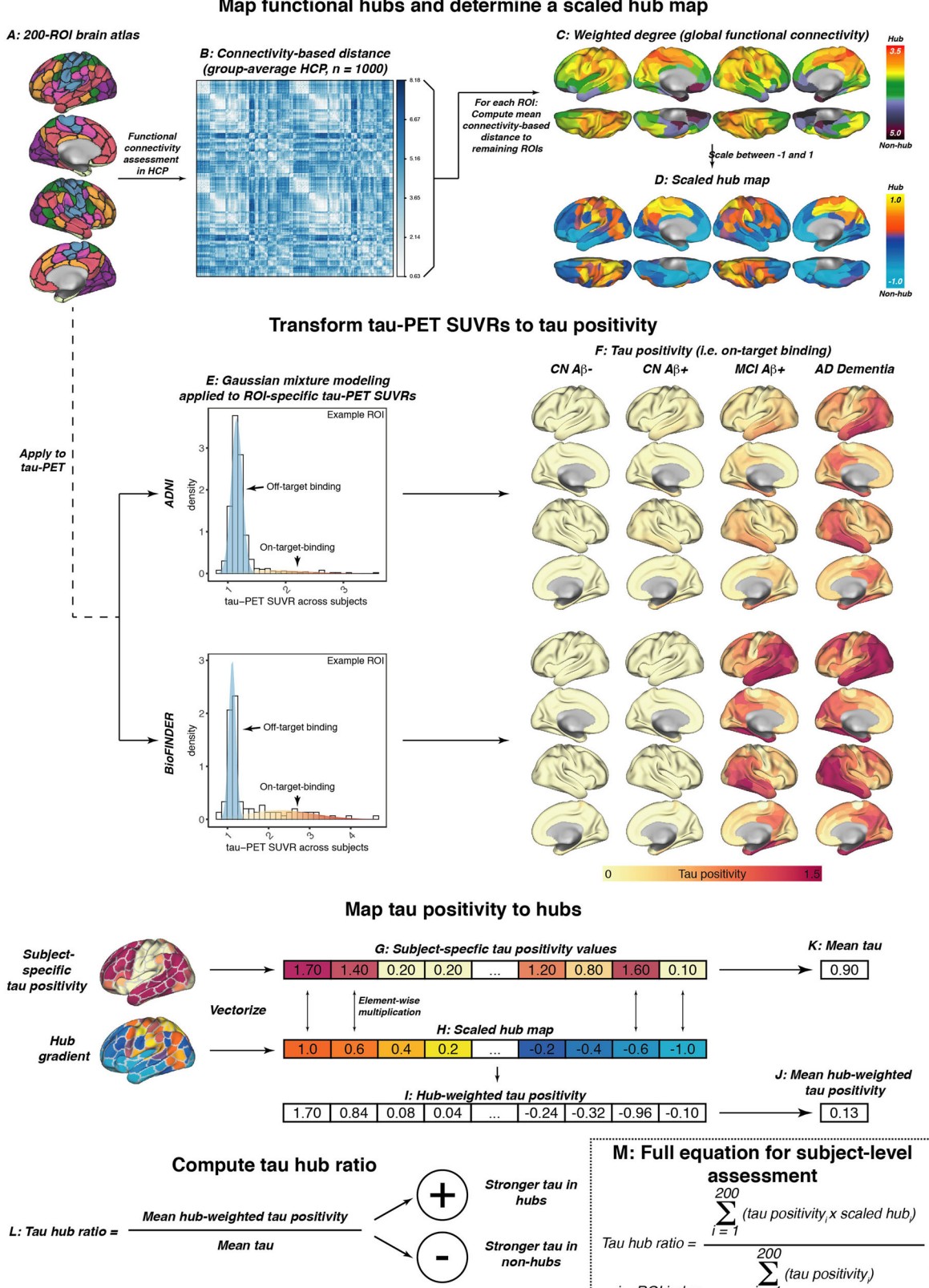

**Map functional hubs and determine a scaled hub map**

*A: 200-ROI brain atlas*

*B: Connectivity-based distance (group-average HCP, n = 1000)*

*C: Weighted degree (global functional connectivity)*

*Functional connectivity assessment in HCP*

*For each ROI: Compute mean connectivity-based distance to remaining ROIs*

*Scale between -1 and 1*

*D: Scaled hub map*

**Transform tau-PET SUVRs to tau positivity**

*E: Gaussian mixture modeling applied to ROI-specific tau-PET SUVRs*

*F: Tau positivity (i.e. on-target binding)*

CN Aβ-    CN Aβ+    MCI Aβ+    AD Dementia

*Apply to tau-PET*

ADNI — Example ROI — Off-target binding — On-target-binding — tau–PET SUVR across subjects

BioFINDER — Example ROI — Off-target binding — On-target-binding — tau–PET SUVR across subjects

0    Tau positivity    1.5

**Map tau positivity to hubs**

*Subject-specific tau positivity*

*G: Subject-specfic tau positivity values*

| 1.70 | 1.40 | 0.20 | 0.20 | ... | 1.20 | 0.80 | 1.60 | 0.10 |

*K: Mean tau* — 0.90

*Vectorize* — *Element-wise multiplication*

*Hub gradient*

*H: Scaled hub map*

| 1.0 | 0.6 | 0.4 | 0.2 | ... | -0.2 | -0.4 | -0.6 | -1.0 |

*I: Hub-weighted tau positivity*

| 1.70 | 0.84 | 0.08 | 0.04 | ... | -0.24 | -0.32 | -0.96 | -0.10 |

*J: Mean hub-weighted tau positivity* — 0.13

**Compute tau hub ratio**

*L: Tau hub ratio =* $\dfrac{\textit{Mean hub-weighted tau positivity}}{\textit{Mean tau}}$

(+) Stronger tau in hubs

(−) Stronger tau in non-hubs

*M: Full equation for subject-level assessment*

$$\text{Tau hub ratio} = \frac{\sum_{i=1}^{200}(\text{tau positivity}_i \times \text{scaled hub}_i)}{\sum_{i=1}^{200}(\text{tau positivity}_i)}$$

i = ROI index

---

$p = 0.002$, Fig. 3B). Together, these results support the hypothesis that younger AD symptom onset is associated with stronger tau pathology in functional hub regions that are strongly interconnected with the rest of the brain.

In an exploratory analysis, we tested whether amyloid-positive ApoE4 non-carriers showed a stronger tau hub ratio, since previous work has suggested that ApoE4-carriage is associated with a more limbic-predominant pattern of tau pathology in AD patients[16], whereas amyloid-positive ApoE4 non-carriers show a more neocortical tau pathology pattern[42]. Supporting this, we found that ApoE4-carriage was associated with a lower tau hub ratio in amyloid-positive (i.e., AD) patients (ADNI/BioFINDER: $F = 4.816/4.404$, Cohen's $d = 0.4/0.3$,

**Fig. 1 | fMRI and tau-PET processing. A** Surface rendering of the 200 region of interest (ROI)-brain atlas, based on which we estimated (**B**) inter-regional functional connectivity-based distance based on 1000 participants of the human connectome project. Specifically, subject-specific connectivity matrices (i.e., Fisher-$z$ transformed correlations of ROI-specific preprocessed fMRI timeseries) were averaged across subjects, thresholded at a density of 30% (i.e., only the strongest 30% of connections were retained) and transformed to connectivity-based distance (i.e., shorter distance = higher connectivity). **C** For each ROI, we determined the average connectivity-based distance to the remaining 199 ROIs as a measure of weighted degree (i.e., global connectivity), which was subsequently rescaled between −1 and 1 to determine a (**D**) continuous mapping of connectivity from hubs to non-hubs. The same brain atlas shown in **A** was applied to tau-PET standardized uptake value ratios (SUVRs), and **E** two-component Gaussian mixture

modeling was applied to transform tau-PET SUVRs to tau positivities, i.e., tau-PET SUVRs that have been cleaned from the off-target binding curve. **F** Surface renderings of group-average tau positivities are shown for each diagnostic group of the ADNI and BioFINDER sample. **G** Subject-specific tau positivity values were subsequently multiplied with the (**H**) scaled hub map to determine (**I**) hub-weighted tau positivity values, giving tau in hub regions a positive weight and tau in non-hub regions a negative weight. **J** Hub-weighted tau positivities and **K** tau positivities were subsequently averaged in order to (**L**) compute the tau hub ratio, i.e., a single numeric index indicating whether an individual tau PET pattern was more pronounced in hub regions (i.e., more positive) or non-hub regions (i.e., more negative), while adjusting for global tau levels. The mathematical equation for determining the tau hub ratio for subject-level data is shown in **M**. Source data are provided as a Source Data file.

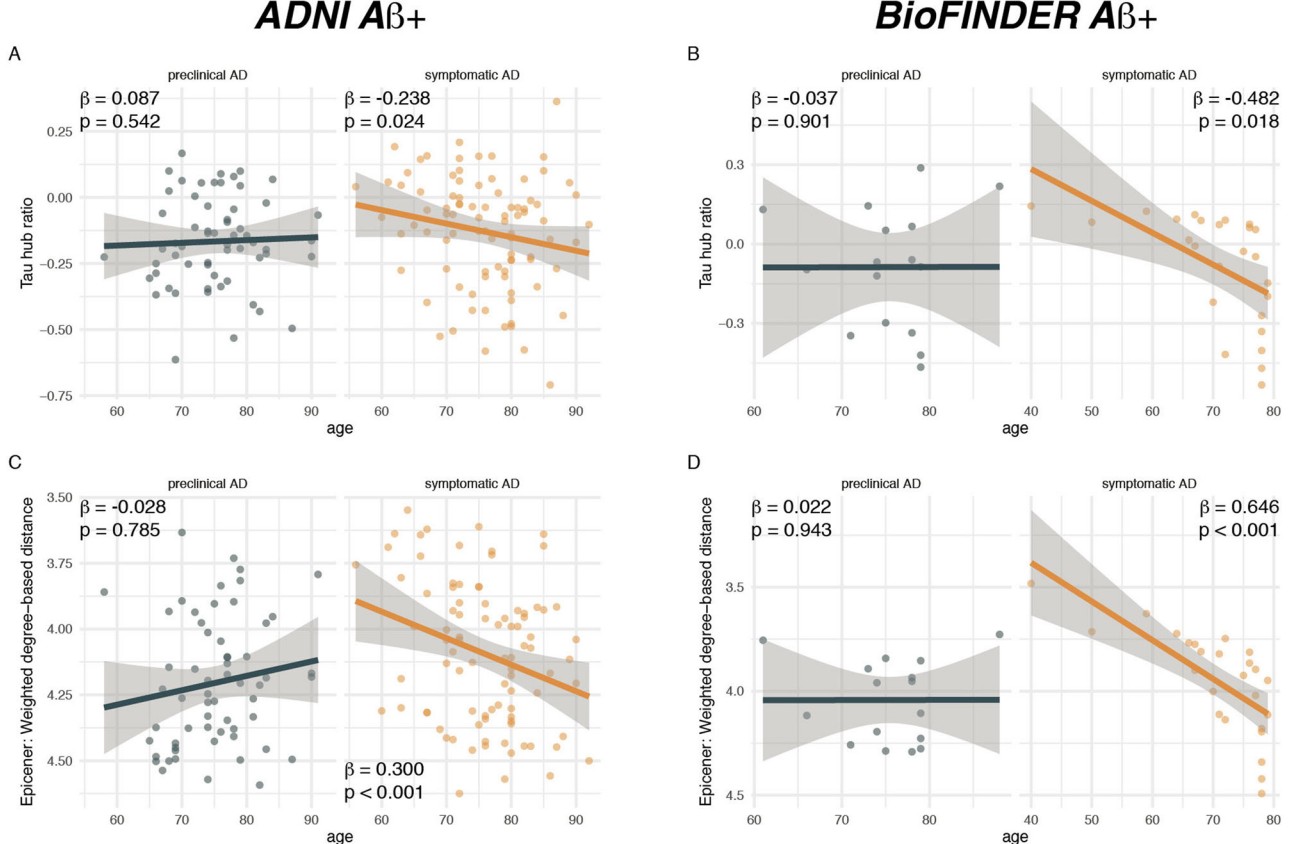

**Fig. 2 | Younger age is associated with a higher hub ratio in symptomatic AD.** Scatterplots illustrating the association between age and the tau hub ratio for Aβ+ subjects of the **A** ADNI ($N = 147$) and **B** BioFINDER sample ($N = 41$), stratified by clinical status. **C, D** illustrate the association between age and the average weighted degree-based distance (i.e., shorter distance = higher connectivity) of tau epi-centers (i.e., 10% of brain regions with highest tau-PET positivity which were

determined on the subject level) in **C** ADNI and **D** BioFINDER. Note that the $y$-axis in **C, D** has been inverted, since lower values mean reflect higher hub-ness. β-values reflect standardized regression weights. All β- and two-sided $p$ values were derived from linear regression controlling for sex, education (and diagnosis in symptomatic AD groups). Linear model fits (i.e., least squares line) are indicated together with 95% confidence intervals. Source data are provided as a Source Data file.

$p = 0.036/0.038$), while controlling for age, sex, education and diagnosis. All above-described results remained consistent when including hippocampal tau-PET in the analyses (see Supplementary Table 2).

### Younger age and a higher tau hub ratio are associated with faster tau accumulation in symptomatic AD

In a next step, we tested whether younger age was associated with a faster tau accumulation rate (i.e., annual change rate in global tau positivity) in patients with symptomatic AD. In line with previous work[3,4], we found that younger age was associated with a faster annual change rate in global tau-PET positivity in patients with symptomatic AD (ADNI: $\beta = -0.284$, $p = 0.009$, Fig. 4B, right panel; BioFINDER: $\beta = -0.836$, $p < 0.001$, Fig. 4C, right panel), using linear regression

controlling for sex, education and diagnosis. As expected, no association between younger age and faster tau accumulation was detected in patients with preclinical AD (ADNI: $\beta = -0.119$, $p = 0.389$, Fig. 4B, left panel; BioFINDER: $\beta = 0.208$, $p = 0.448$, Fig. 4C, left panel), suggesting that faster tau accumulation or tau accumulation in general at younger age is specific for subjects with symptomatic AD. Next, we tested whether a higher tau hub ratio, indicating a more hub-like tau-PET deposition pattern, was associated with faster global tau accumulation. This analysis was motivated by our previous findings, showing that tau spreads from circumscribed epicenters to connected brain regions[33,34]. Thus, tau pathology in a globally connected hub region should result in more widespread tau spreading and faster tau accumulation. In contrast, tau in only weakly interconnected non-

## ADNI - symptomatic AD
### (n=77)

A

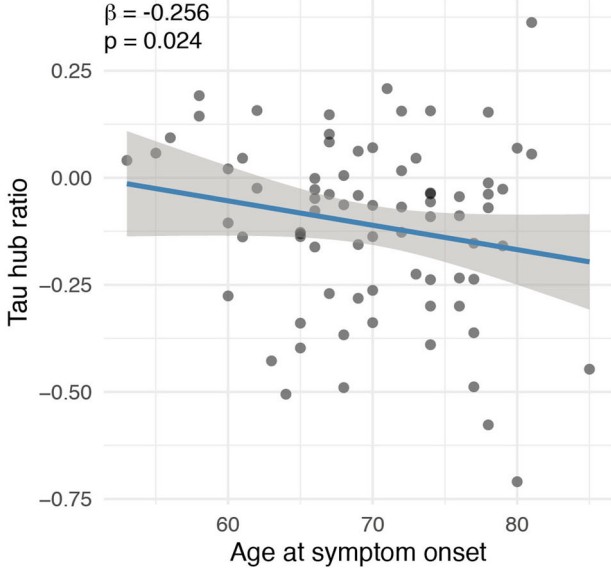

β = -0.256
p = 0.024

B

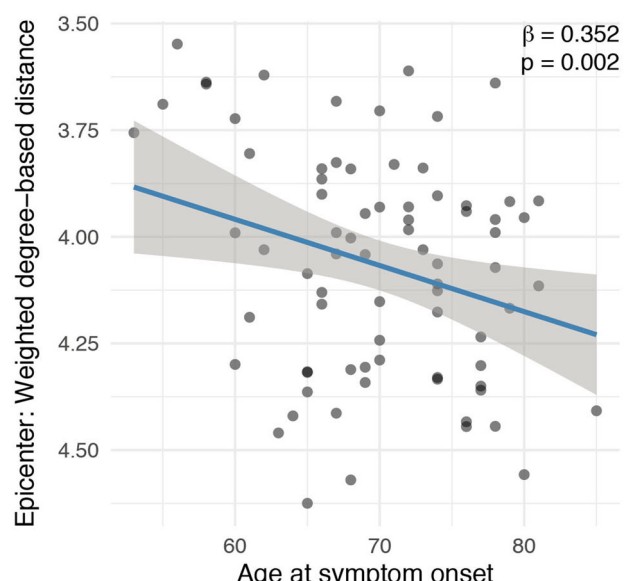

β = 0.352
p = 0.002

**Fig. 3 | Younger symptom onset is associated with a higher tau hub ratio.** Scatterplots illustrating the association between (**A**) actual age of onset and the tau hub ratio for *n* = 77 symptomatic AD subjects of the ADNI sample as well as between (**B**) age at symptom onset and global connectivity-based distance of the tau epicenters. β-values reflect standardized regression weights. All β-and two-sided *p* values were derived from linear regression controlling for sex, education and diagnosis and the time difference between age of symptom onset and the first tau-PET scan. Linear model fits (i.e., least squares line) are indicated together with 95% confidence intervals. Source data are provided as a Source Data file.

hub regions should result in slower spreading and tau accumulation (see Fig. 4A for an example illustration). Supporting this, we found that a higher tau hub ratio at baseline predicted faster subsequent tau accumulation in patients with symptomatic AD (ADNI: β = 0.347, *p* = 0.006, Fig. 4D, right panel; BioFINDER: β = 0.668, *p* = 0.002, Fig. 4E, right panel), controlling for sex, education and diagnosis. Exact tests using 1000 beta values from null-model tau hub ratios that

were generated using shuffled connectomes as a reference confirmed these results (ADNI: *p* = 0.010; BioFINDER: *p* = 0.018). This result remained consistent when additionally controlling for age (ADNI: β = 0.259, *p* = 0.009; BioFINDER: β = 0.311, *p* = 0.026), suggesting a unique contribution of the tau hub ratio to faster tau accumulation. In patients with preclinical AD, we also found a significant association between a higher tau hub ratio at baseline and a faster subsequent tau accumulation rate in ADNI (β = 0.347, *p* = 0.006, Fig. 4D, left panel), controlling for sex and education, however this finding could not be replicated in BioFINDER (β = 0.199, *p* = 0.469, Fig. 4E, left panel). In summary, these results suggest that younger age and a more hub-like tau-PET retention pattern are associated with a faster accumulation of tau pathology in patients with symptomatic AD. Again, these results remained consistent when including hippocampal tau-PET (see Supplementary Table 2).

To further confirm that it is indeed the spatial distribution of tau pathology in hubs and not the overall severity of tau pathology that drives faster tau accumulation and earlier disease manifestation, we performed additional confirmatory analyses. Specifically, we conducted a sliding-window analysis from low to high global tau-levels analysis in Aβ+ subjects of ADNI and BioFINDER, and assessed for each window the mean annual tau-PET change and age, stratified by a high vs. low tau hub ratio (i.e., median split) or by high vs. low epicenter connectivity-based distance. We found significant interactions of global tau-PET and the tau hub ratio on the annual rate of tau accumulation (Supplementary Fig. 1A, B), where a higher tau hub ratio was associated with faster tau accumulation at higher baseline tau-PET levels. Similarly, we found an interaction between global tau-PET and the tau hub ratio on age in symptomatic AD patients, where a higher tau hub ratio was associated with younger age at higher tau-PET levels. These analyses support the view that a higher tau hub ratio is associated with faster tau accumulation at a given level of tau pathology in AD, as well as with younger age in symptomatic AD. Similarly, we found that at a given level of global tau-PET, more globally connected epicenters were associated with faster subsequent tau accumulation. Analyses are summarized in Supplementary Fig. 2A–F.

### A higher tau hub ratio mediates the association between younger age and faster tau accumulation in symptomatic AD

Lastly, we assessed in symptomatic AD patients whether the effect of younger age on faster tau accumulation rates was mediated via stronger tau pathology in hub regions, i.e., a higher tau hub ratio. Using bootstrapped mediation analyses with 10,000 iterations, we found a significant mediation effect both in ADNI (β = −0.065, 95% CI [−0.153; 0.00], *p* = 0.039) and BioFINDER (β = −0.149, 95% CI [−0.371; 0.00], *p* = 0.046), controlling for sex, education and diagnosis. Mediation analyses are summarized in Fig. 5A, suggesting that a higher tau hub ratio mediates the association between younger age and faster tau accumulation rates in patients with symptomatic AD. These data remained consistent when also including hippocampal tau-PET (ADNI: β = −0.069, 95% CI [−0.164; −0.01], *p* = 0.024; BioFINDER: β = −0.149, 95% CI [−0.366; −0.01], *p* = 0.033). Using longitudinal cognitive data available in symptomatic AD patients of the ADNI sample, we could further show that faster annual tau accumulation was associated with a faster decline in a memory composite (i.e., ADNI-MEM, β = −0.322, *p* = 0.004, Fig. 5B, controlling for age, sex and education), suggesting that faster tau hub ratio-dependent accumulation of tau pathology may drive faster clinical deterioration in younger patients with clinical AD. Again, this association remained consistent when including hippocampal tau-PET (β = −0.315, *p* = 0.004). Confirming these analyses using the sliding-window approach introduced above, we found an interaction between baseline global tau-PET and the tau hub ratio on annual changes in ADNI-MEM, where a higher tau hub ratio was associated with faster cognitive decline at higher overall tau levels.

                                                                                                                                  

# Hypothetical model of connectivity-mediated tau spreading

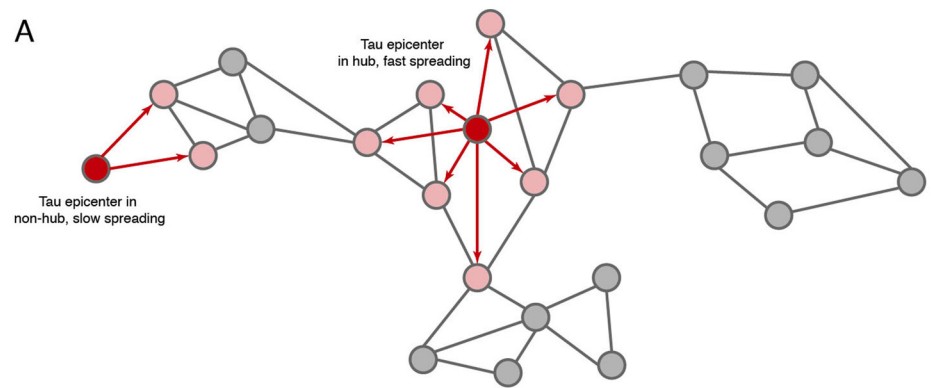

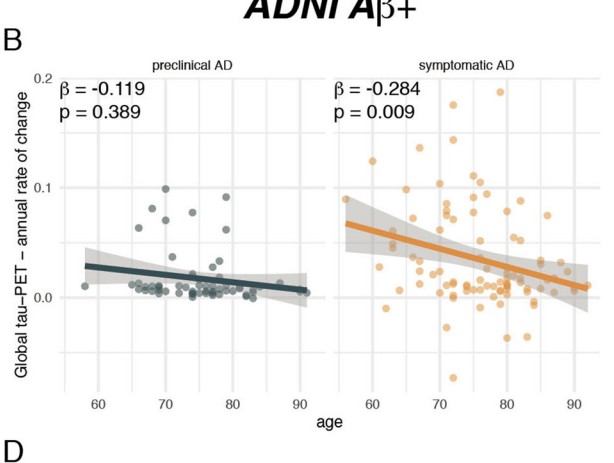

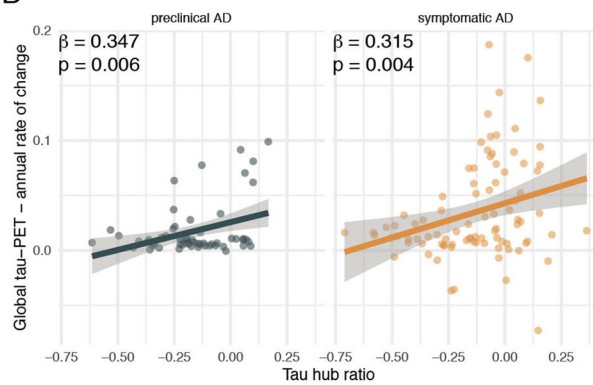

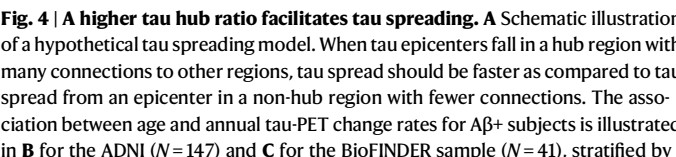

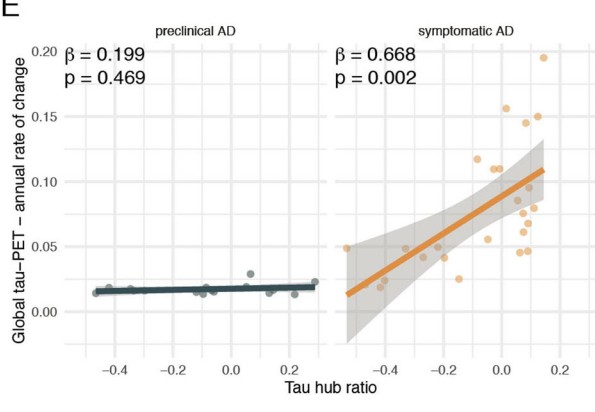

**Fig. 4 | A higher tau hub ratio facilitates tau spreading. A** Schematic illustration of a hypothetical tau spreading model. When tau epicenters fall in a hub region with many connections to other regions, tau spread should be faster as compared to tau spread from an epicenter in a non-hub region with fewer connections. The association between age and annual tau-PET change rates for Aβ+ subjects is illustrated in **B** for the ADNI ($N = 147$) and **C** for the BioFINDER sample ($N = 41$), stratified by clinical status. Scatterplots showing the association the tau hub ratio and annual tau accumulation rates in **D** ADNI and **E** BioFINDER. β-values reflect standardized regression weights. All β-and two-sided *p* values were derived from linear regression controlling for sex, education (and diagnosis in symptomatic AD groups). Linear model fits (i.e., least squares line) are indicated together with 95% confidence intervals. Source data are provided as a Source Data file.

## Discussion

The main findings of the current study were first, that younger patients with symptomatic AD show a more hub-like pattern of PET-assessed tau deposition, i.e., stronger tau in highly connected brain regions which are critical for domain-general cognition[22] and which we have previously shown to be central for maintaining cognitive performance in AD[25–27,43]. Importantly, these associations were not detected for amyloid-positive, yet asymptomatic individuals (i.e., preclinical AD), in whom tau pathology was mostly absent. This supports previous findings of tau-pathology being a key driver of symptomatic disease manifestation in AD[1,14,15,44], while adding important new evidence that stronger tau deposition in globally connected hubs is associated with earlier AD symptom manifestation. Second, we found that younger age and a stronger load of tau pathology in globally connected hubs predicted faster subsequent tau accumulation rates in symptomatic AD patients which was associated with faster decline in memory performance. Connectivity is assumed to be a key mediator of tau spreading[31,33–35], hence these results favor the hypothesis that tau

**A**

***Mediation analyses in patients with symptomatic AD***

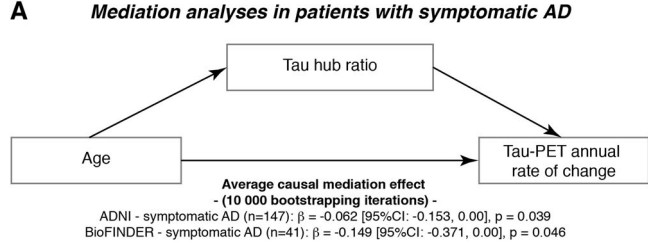

Average causal mediation effect
- (10 000 bootstrapping iterations) -
ADNI - symptomatic AD (n=147): β = -0.062 [95%CI: -0.153, 0.00], p = 0.039
BioFINDER - symptomatic AD (n=41): β = -0.149 [95%CI: -0.371, 0.00], p = 0.046

**B**

***ADNI - symptomatic AD (n=87)***

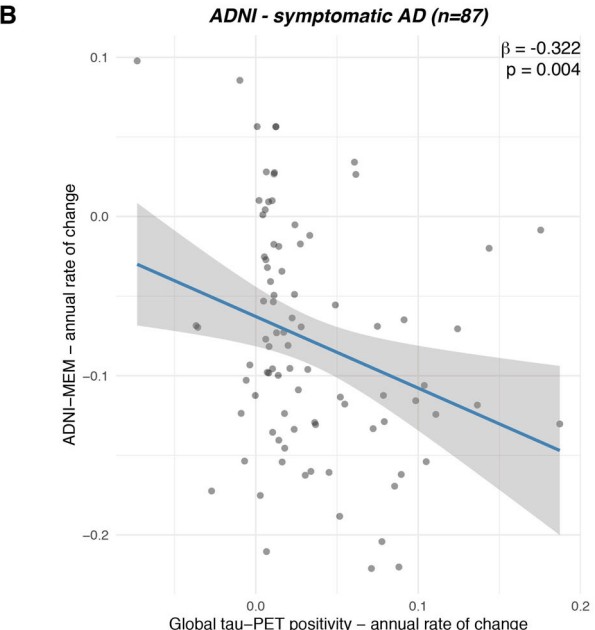

β = -0.322
p = 0.004

**Fig. 5 | The association between younger age and faster tau accumulation is mediated by a higher tau hub ratio. A** Mediation analysis in patients with symptomatic AD (ADNI: $N = 87$, BioFINDER: $N = 25$), showing that the association between younger age and faster annual tau accumulation rates is mediated by tau hub ratio. Mediation effects (i.e., β-values, 95% confidence intervals [CI] and uncorrected two-sided *p* values) were determined based on bootstrapping with 10,000 iterations, while all paths were controlled for age, sex and education. **B** In the ADNI symptomatic AD sample with available longitudinal cognition on a memory composite (i.e., ADNI-MEM, $n = 87$), we plotted the association between tau-PET accumulation rates (*x*-axis) and cognitive change rates (*y*-axis). β-values reflect standardized regression weights. β-and two-sided *p* values were derived from linear regression, controlling for age, sex and education. Linear model fits (i.e., least squares line) are indicated together with 95% confidence intervals. Source data are provided as a Source Data file.

pathology in globally connected hubs early in the disease course may lead to more widespread propagation of tau pathology across connected brain regions, thereby driving faster cognitive deterioration in patients with earlier symptomatic AD manifestation[14,15,30]. To integrate our findings on how age, tau deposition patterns and tau accumulation rates are inter-related, we show that the association between younger age and faster tau accumulation in symptomatic AD patients is mediated by stronger tau pathology in hub regions. Together, the current independently validated findings provide key evidence that a predilection of tau pathology deposition toward globally connected brain hubs that are crucial for cognition[22] may determine earlier symptomatic disease manifestation and accelerate connectivity-mediated tau spreading.

For our first finding, we show that patients with younger symptomatic manifestation of AD show stronger tau pathology in globally connected hubs in the fronto-parietal association cortex compared to

inferior temporal non-hubs (Fig. 2). This finding aligns well with previous post-mortem[10] and in vivo tau-PET evidence[12,16-19], showing that younger symptomatic AD onset is associated with a more neocortical limbic sparing tau pathology pattern, whereas older age of symptom onset is associated with a limbic-predominant tau pathology pattern[10]. Similarly, previous studies found that younger symptomatic AD onset is associated with stronger fronto-parietal gray-matter atrophy[5], hypoperfusion and glucose hypometabolism[45,46], which have been shown to be closely associated with tau pathology in AD[13]. Together, this suggests that the spatial pattern rather than the mere extent of tau pathology may determine the likelihood, type and aggressiveness of symptom manifestation in AD[1,12,47]. The current findings critically extend these previous results by showing that younger symptomatic AD manifestation is specifically associated with a tau pathology pattern that resembles hub regions that are critical for domain general cognitive functioning[22,40]. Supporting a role of hubs for cognition in AD, we found previously that higher global connectivity fronto-parietal control network hubs such as the left frontal cortex is associated with a more efficient (i.e., small-world) global brain network topology[48] and attenuated effects of posterior-cingulate glucose hypometabolism[25], hippocampal atrophy[43,49,50] or entorhinal tau pathology on cognitive performance[27]. This suggests that maintaining the integrity of hub regions may promote resilience to the impact of AD pathology on cognition[49]. In turn, damage to hub regions has been shown to diminish the efficient communication between brain networks[51], and clinical studies in patients with cerebrovascular disease have shown that focal lesions in hubs cause brain-wide network disruptions[52,53] and stronger impairment than lesions in non-hub regions[54]. Tau pathology has been shown to disrupt neuronal activity and connectivity in both preclinical[29] and clinical studies[55-58], hence early occurrence of tau pathology in hub regions may lead to an earlier impairment of global brain network function[51] and thus earlier onset of cognitive impairment. Here, it will be an important next step to address whether higher tau pathology in hubs in fact drives a disruption of hub connectivity and thus an earlier symptom onset.

For our second finding, we could show that younger age and a stronger tau hub ratio are associated with faster tau accumulation in symptomatic AD. This is a critical extension of previous preclinical and clinical work, showing that brain connectivity mediates tau spreading[33-35,59]. Specifically, studies in mice and neuronal cell-cultures found that tau spreads trans-synaptically, where higher synaptic activity facilitates tau spreading[31,32,59]. Similarly, we and others reported in combined MRI and tau-PET studies in AD patients that seed-based connectivity of tau harboring epicenters predict tau spreading patterns, where tau spreads preferentially from epicenters to connected regions[2,33-37]. Thus, a tau harboring hub region with widespread connections may potentiate tau spreading to connected regions compared to a non-hub with fewer connections (i.e., as illustrated in Fig. 4A). Supporting this, we found that younger symptomatic AD patients had regions of highest tau pathology (i.e., epicenters) in more globally connected brain regions and that a stronger tau hub ratio was associated with faster tau accumulation. Importantly, our findings also offer a mechanistic explanation for previous results on faster tau accumulation rates in younger AD patients[3,4], which is further supported by our mediation analyses showing that the association between younger age and faster tau accumulation rates is mediated by a stronger tau hub ratio in symptomatic AD. Importantly, faster tau accumulation was associated with faster decline in memory performance, supporting the view that faster tau accumulation drives the rapid clinical progression in AD patients with earlier symptom manifestation[6-8,30]. Yet, we caution that these results on cognition were restricted to ADNI and warrant further validation in future studies. Nevertheless, our results suggest that hubs may play an important role as distributors of tau pathology, driving more widespread tau propagation and thus clinical deterioration. Of note, we also found an

association between the tau hub ratio and faster tau accumulation in preclinical AD patients in ADNI but not in BioFINDER, potentially since preclinical AD patients in ADNI already show slightly more advanced tau pathology than in BioFINDER (see Table 1) and may therefore be at higher risk of subsequent tau spreading[38].

For interpreting the results of the current study, several caveats should be taken into account. First, Flortaucipir tau-PET shows considerable off-target binding in subcortical regions including the hippocampus and basal ganglia[60]. To address this, we excluded regions that are known to be severely affected by unspecific binding for our main analyses (e.g., hippocampus, basal ganglia) and further minimized any influence of Flortaucipir off-target binding by transforming tau-PET SUVRs to tau positivities using a Gaussian mixture modeling approach that has been previously applied by amyloid- and tau-PET studies to separate target- from unspecific binding[33,36,61]. In an exploratory approach, we also included the hippocampus using a pre-established hippocampal mask[41] that excludes the choroid plexus (i.e., the main source of off-target binding), yielding fully consistent results with our main analyses (see Supplementary Table 2). Still, it is possible that unspecific binding influences our results, hence our results await further replication using second-generation tau-PET data with a better off-target binding profile. Second, the current study included only include a small number of patients that would be clinically considered as early-onset AD (i.e., below the age of 65, ADNI/BioFINDER, $n = 8/4$). Therefore, the current study is mostly limited to the variability of tau deposition in the age range of typical late-onset AD. To specifically replicate the role of tau pathology in hubs in subjects with a clinical disease onset before the age of 65, we encourage future studies to assess whether our results can be replicated in dedicated early-onset AD datasets such as the LEADs cohort (https://clinicaltrials.gov/ct2/show/NCT03507257), once sufficient data become available. Third, the current study exclusively used weighted degree (i.e., global connectivity) as a measure of hub-ness[62,63], which yielded similar hub patterns as in previous studies[20,63]. We specifically selected this measure, since it is relatively easy to interpret (i.e., strength of connectedness to the rest of the brain) and well-suited to test the effect of connectivity on tau spreading. We are, however, aware of the breadth of graph-theoretical measures that have been proposed to quantify the hub-ness of a given brain region, yet many of these measures are relatively abstract (e.g., betweenness-centrality, participation coefficient), and often highly intercorrelated[21,64]. To avoid introducing additional and potentially redundant hub measures to an already complex set of analyses, we thus refrained from repeating the analyses with alternative measures of hub-ness. Fourth, other modulating factors such as reserve and resilience may influence symptom onset and progression rates in AD[65]. To account for inter-individual differences in reserve and resilience, all analyses were statistically controlled for years of education, i.e., a well-established proxy of reserve and resilience in AD[65]. However, we cannot exclude that inter-individual differences in reserve/resilience may further modulate symptom onset and progression rates. Lastly, we would like to highlight that we did not perform partial volume correction since longitudinal T1-weighted MRI data was not consistently available and since previous studies showed that longitudinal tau-PET changes can also be reliably detected without partial-volume correction[66,67].

In conclusion, our independently validated results provide novel evidence that younger symptomatic AD patients show stronger tau pathology in globally connected hubs, which may drive faster tau spreading and accelerated cognitive decline. This suggests that earlier symptom manifestation is not driven by specific pathophysiological characteristics[11], but rather by a tau distribution pattern[10,12] that preferentially targets brain hubs important for cognitive function[22,25–27,49,50]. These results converge well with previous findings, showing that heterogeneous tau distribution patterns are associated with heterogeneous clinical trajectories, variable symptom onset and

disease progression[1,2,10,12,14,15,33,47]. Here, it will be a key future goal to identify potential determinants of spatially variable tau pathology onset, such as differential gene expression patterns[68–70], pre-existing tau pathology (e.g., related to traumatic brain injury)[71,72], or premorbid differences and/or heterogeneity in brain network architecture[73]. Knowledge about drivers of tau onset, heterogeneous tau spreading patterns and clinical trajectories may become important to facilitate precision-medicine prediction of cognitive and pathological progression, as well as for patient stratification in clinical trials[14,74].

## Methods

### Participants—ADNI
We included 240 participants of the ADNI (ClinicalTrials.gov Identifier: NCT02854033) database. Inclusion criteria were availability of longitudinal [18]F-Flortaucipir tau-PET as well as baseline [18]F-Florbetapir or [18]F-Florbetaben amyloid-PET, T1 MPRAGE structural MRI and demographic data. All baseline imaging had to be collected within 12 months. Aβ-status was assessed based on whole-cerebellum normalized global amyloid-PET SUVR, using pre-established protocols and cut-points (global AV45 SUVR > 1.11; global FBB SUVR > 1.08)[75]. Global amyloid-PET SUVRs were further transformed to the Centiloid scale to allow pooling across tracers[76]. Clinical status was assessed by ADNI, categorizing subjects as CN (MMSE > 24, CDR = 0, non-depressed), MCI (MMSE > 24, CDR = 0.5, objective memory-impairment on education adjusted Wechsler Memory Scale II, preserved activities of daily living), or demented (MMSE 20–26, CDR > 0.5, NINCDS/ADRDA criteria for probable AD). Amyloid-positive ($n = 149$) individuals were grouped as preclinical AD (i.e., CN, $n = 60$) or AD clinical syndrome (i.e., MCI [$n = 57$] and Dementia [$n = 32$]). Ethical approval was obtained by the ADNI investigators at each participating site, all participants provided written informed consent. All study relevant covariates are described in Table 1.

### Participants—BioFINDER
As a replication sample, we included 57 BioFINDER (ClinicalTrials.gov Identifier: NCT01208675) participants with available Aβ-status, structural MRI and longitudinal [18]F-Flortaucipir tau-PET. Baseline Aβ-status was determined using [18]F-Flutemetamol-PET as described previously[61], applying a pons-normalized global SUVR cut-off >0.575[77]. The Alzheimer's continuum was covered by 16 CN Aβ+, 7 MCI Aβ+ and 18 AD dementia subjects vs. 16 CN Aβ- subjects as controls. BioFINDER inclusion and exclusion criteria as well as diagnostic criteria have been described previously[78]. As for ADNI, amyloid-positive individuals were grouped as preclinical AD (i.e., CN) or AD clinical syndrome (i.e., MCI and Dementia). All participants provided written informed consent prior to study inclusion. Ethical approval was provided by the ethics committee at Lund University, Sweden. Imaging procedures were approved by the Radiation protection committee at Skåne University Hospital and by the Swedish Medical Products Agency. All study relevant covariates are described in Table 1.

### MRI and PET acquisition and preprocessing in ADNI
All ADNI structural MRI data were obtained on 3T scanners, using a 3D T1-weighted MPRAGE sequence with 1 mm isotropic voxel-size (TR = 2300 ms, for parameter details see: https://adni.loni.usc.edu/wp-content/uploads/2017/07/ADNI3-MRI-protocols.pdf). Subject-level 3T resting-state fMRI data was included for 77 individuals with an available resting-state fMRI scan within 1 year of the baseline tau-PET scan (3T EPI sequence, TR/TE/flip angle: 3000 ms/30 ms/90°, 200 volumes, 3.4 mm isotropic voxel-resolution). PET data were acquired at standardized time-intervals post injection of [18]F-labeled tracers (Flortaucipir: 6 × 5 min time-frames, 75–105 min post injection; Florbetapir: 4 × 5 min time-frames, 50–70 min post injection; Florbetaben: 4 × 5 min time-frames, 90–110 min post injection; further details can be found at http://adni.loni.usc.edu/methods/pet-analysis-method/pet-

analysis/). Dynamically acquired images were realigned and averaged to obtain single Flortaucipir/Florbetapir/Florbetaben images. Structural T1 MRI images were normalized to Montreal Neurological Institute (MNI) standard space using Advanced Normalization Tools (ANTs)[79]. PET images were then coregistered to native-space T1 images, and subsequently normalized to MNI space by applying the ANTs normalization parameters. From each spatially normalized PET image, we then extracted mean values of 200 ROIs covering the entire neocortex, using an established brain parcellation (see Fig. 1A)[39]. For exploratory reasons, we added two pre-established hippocampal ROIs (i.e., left vs. right) to this brain parcellation, which exclude the choroid plexus to allow quantification of hippocampal tau-PET while minimizing the bias included by off-target binding in the choroid plexus[41]. To tailor all ROIs to the current sample, we additionally masked all ROIs with a group-specific gray matter mask binarized at a probability of 0.3.

For 77 subjects with available resting-state fMRI, we first applied motion correction (i.e., realignment), regressed out the mean signal from the white matter, cerebrospinal fluid as well as the 6 motion parameters that were estimated during realignment (i.e., 3 translations and 3 rotations). Next we applied detrending, band-pass filtering (0.01–0.08 Hz) and despiking. To further eliminate motion artifacts, we performed scrubbing, i.e., removal of high-motion frames as defined by exceeding 0.5 mm framewise displacement. Specifically, high-motion volumes together with one preceding and two subsequent volumes were replaced with zero-padded volumes to eliminate high-motion volumes but keep the number of volumes consistent across subjects. Lastly, the preprocessed resting-state fMRI images were spatially normalized to MNI space by (1) coregistration to the baseline T1-weighted images followed by applying the ANTs-derived non-linear transformation parameters.

### MRI and PET acquisition and preprocessing in BioFINDER
In BioFINDER, 1 mm isotropic T1-weighted MPRAGE (TR = 1900 ms) and Fluid-attenuated inversion recovery (FLAIR; $0.7 \times 0.7 \times 5 \text{ mm}^3$ voxel size, 23 slices, TR = 9000 ms) images were acquired for all participants on a 3T Siemens Skyra scanner (Siemens Medical Solutions, Erlangen, Germany). Tau-PET was recorded 80–100 min after bolus injection of [18]F-Flortaucipir on a GE Discovery 690 PET scanner (General Electric Medical Systems, Milwaukee, WI, USA). Imaging data were processed by the BioFINDER imaging core using a pipeline developed at Lund University that was described previously[80]. In brief, MRIs were skull stripped using the combined MPRAGE and FLAIR data, segmented into gray and white matter and normalized to MNI space. PET images were attenuation corrected, motion corrected, summed and coregistered to the MRIs. In line with ADNI data, standardized uptake value ratio (SUVR) data were calculated using an inferior cerebellar gray matter as reference region. From spatially normalized PET, we extracted mean values of 200 ROIs included in the gray matter masked brain parcellation (Fig. 1A)[39]. Usage of an alternative reference region (i.e., eroded white matter) yielded consistent results with the analyses reported in the manuscript. As for the ADNI sample, we also extracted tau-PET SUVRs from two hippocampal ROIs that exclude the choroid plexus to minimize the effects of off-target binding[41].

### Assessment of functional connectivity and connectivity-based distance
To determine a map of functional hubs, we downloaded spatially normalized (i.e., to MNI space) minimally preprocessed 3T resting-state fMRI images from 1000 subjects of the HCP. We further applied detrending, band-pass filtering (0.01–0.08 Hz), despiking and motion scrubbing to the HCP resting-state data. To further reduce motion artifacts, we performed scrubbing, i.e., removal of high-motion frames as defined by exceeding 0.5 mm framewise displacement, where high-motion volumes together with one preceding and two subsequent volumes were replaced with zero-padded volumes to

eliminate high-motion volumes but keep the number of volumes consistent across subjects. We applied the Schaefer 200 ROI atlas plus the two hippocampal ROIs and extracted mean ROI timeseries, which were cross-correlated and Fisher-z transformed in order to determine functional connectivity for each subject. The 1000 subject-specific functional connectivity matrices were subsequently averaged and thresholded at a density of 30%. The resulting thresholded functional connectivity matrix was then converted to distance (i.e., shortest path-length between ROIs, Fig. 1B)[64] and we obtained the average distance of a given ROI to the remaining ROIs as a measure of node degree or global connectivity (Fig. 1C). The resulting connectivity map was then scaled between −1 (i.e., high average distance, non-hub) and 1 (i.e., low average distance, hub) to determine a scaled hub map that was later applied to tau-PET data (see below, Fig. 1D). This scaled hub map was determined either including or excluding the two hippocampal ROIs. Note that we did not perform global signal regression due to some controversies about potential bias introduced by this preprocessing step[81]. However, when reanalyzing the data with global signal regression, all results presented in this manuscript remained consistent.

For ADNI participants with available resting-state fMRI data, we used the same above-described approach in order to determine subject-level scaled hub maps.

### Transforming tau-PET SUVRs to tau positivity and longitudinal tau-PET change
[18]F-Flortaucipir tau-PET shows considerable off-target binding across the brain, causing signal in brain regions which do not harbor pathological tau[82]. To minimize the impact of off-target binding, we performed our previously established approach that applies Gaussian-mixture modeling to PET data in order to separate target from off-target binding[33,36,61]. The rationale is that many AD individuals and healthy controls should not show pathological tau in most brain regions, hence pathological tau-PET signal should show a skewed distribution. In contrast, off-target binding should be unspecific and thus show a normal distribution. A mixture of target and off-target signal should thus result in a bimodal distribution which can be separated using Gaussian-mixture modeling. To separate target from off-target tau-PET signal, we extracted tau-PET SUVRs of the 200 ROIs included in the brain atlas (Fig. 1A)[39] plus the two hippocampal ROIs[41] and applied Gaussian-mixture modeling to ROI-specific tau-PET values across the ADNI or BioFINDER sample (Fig. 1E). We fitted one- and two-component models and determined the model with the best fit using Akaike's information criterion, revealing a better two-component fit for all ROIs within both samples. For each subject and ROI, we then determined the probability of falling on the right-most distribution of the two fitted Gaussians. Since this right-most distribution likely reflects abnormal Flortaucipir tau-PET signal, the probability score expresses the proximity of a subject to the pathological distribution, which can thus be interpreted as a probabilistic measure of tau positivity. This probabilistic measure was then multiplied with the actual tau-PET SUVR in order to obtain an SUVR score that was cleaned from off-target signal (Fig. 1F). To determine longitudinal tau-PET changes, we fitted linear mixed models with tau-PET positivity scores as the dependent variable and time (i.e., years from baseline) as the independent variable, controlling for random slope and intercept[83]. From these models, we derived a slope estimate for tau-PET positivity change per year for each subject and ROI. A global tau accumulation rate was calculated as the mean of annual tau-PET positivity change across all ROIs either in- or excluding the two hippocampal ROIs.

### Mapping tau-PET patterns to the scaled hub map
In order to determine whether individual tau-PET patterns resembled a hub-like or non-hub-like pattern (Fig. 1C), we mapped tau patterns to

the connectivity-based scaled hub map (Fig. 1D) that ranges between 1 (i.e., hubs) and −1 (i.e., non-hubs). To this end, we vectorized ROI-based subject-specific tau positivity values (Fig. 1G) and the hub gradient (Fig. 1H) and performed element wise multiplication to determine hub-weighted positivity values (Fig. 1I). By multiplying tau positivity values with the scaled hub map (Fig. 1D), tau in hub regions will be assigned a positive weight, while tau in non-hub regions will be assigned a negative weight. We then assessed the average hub-weighted tau positivity (Fig. 1J), where positive values indicate a stronger tau in hub regions, whereas negative values indicate stronger tau in non-hub regions. In addition, we computed the global mean tau positivity (Fig. 1K), which was used to adjust the mean hub-weighted tau positivity for global tau, yielding the subject-specific tau hub ratio (Fig. 1L, M). Equation 1 summarizes the approach to map an individual tau-PET pattern to the scaled hub map, where $i$ is the index of a given ROI from 1 to 200:

$$\text{tau hub ratio} = \frac{\sum_i^{200}(\text{tau positivity}_i \times \text{scaled hub}_i)}{\sum_i^{200}(\text{tau positivity}_i)} \qquad (1)$$

Note that the same equation was used in ADNI participants with available resting-state fMRI data, using subject-level connectivity data.

## Statistical analysis

For both samples, baseline demographics were compared between diagnostic groups within the ADNI and the BioFINDER sample using chi-squared tests for categorical variables and ANOVAs for continuous variables. In a first step, we performed linear regression to test the association between age and the tau hub ratio, stratified by clinical status (i.e., preclinical vs. symptomatic AD). Models were controlled for sex and education in preclinical AD, and additionally for diagnosis (i.e., MCI or dementia) in symptomatic AD groups. Next, we determined the epicenters of tau pathology for each subject (i.e., 10% of ROIs with highest baseline tau-PET positivity), and computed the mean connectivity-based distance within the epicenter ROIs as a measure of epicenter hub-ness. Using linear regression stratified by clinical status, we then tested whether younger age was associated with a shorter global connectivity-based distance (i.e., higher hub-ness) of tau epicenters, controlling for sex and education (and diagnosis in symptomatic AD). In symptomatic AD subjects of the ADNI cohort, the above-described models were re-ran with actual ages of symptom onset which were available in $n = 77$ symptomatic ADNI subjects, while additionally controlling for the time difference between the actual age of onset and the baseline tau-PET scan. In a next step, we tested whether younger age was associated with a faster annual tau accumulation rate (i.e., average annual rate of change in tau-PET positivity across all 200 cortical ROIs), using linear regression controlling for sex, education (and diagnosis in symptomatic AD). Equivalent models were run using the tau hub ratio as a predictor of annual tau accumulation rates to test whether a stronger tau in hubs is associated with faster tau spreading. All above models were additionally run 1000 times using tau hub ratios which were created using randomly shuffled connectivity values as null-models. The true beta values determined on the models using the actual tau hub ratio were then compared against 1000 beta values derived from shuffled tau hub ratio models using an exact test.

To further test whether the deposition pattern rather than the overall severity of tau was associated with younger age and a faster rate of tau accumulation in symptomatic AD, we performed a sliding-window analysis. Specifically, we sorted our sample from low to high global tau-PET levels (i.e., the average tau-PET positivity across the entire cortex). We then used a window size of 30% of the entire Aβ+ sample, which was shifted in steps of 3 subjects from low to high global tau-PET levels. Within each window, we quantified the mean annual global tau-PET change for subjects with a high (i.e., >median) tau hub

ratio and in subjects with a low (i.e., <median) tau hub ratio. With this analysis approach, we were able to test whether at a given level of baseline tau levels, a higher tau hub ratio at baseline was associated with faster subsequent tau accumulation, i.e., to specifically determine whether the topographical distribution and not the overall severity of tau in hubs is associated with accelerated tau accumulation. To statistically test this hypothesis, we used linear regression to assess the interaction between baseline global tau-PET positivity (i.e., determined within each sliding-window sample) and the tau hub ratio on annual global tau-PET increases, as well as on age.

Lastly, we conducted bootstrapped mediation analyses using the R-package mediation (Version 4.5.0) in order to test whether the association between younger age and faster tau accumulation rates in symptomatic AD was mediated by a higher tau hub ratio. All paths of the mediation model were controlled for sex, education and diagnosis. Significance of the mediation effect was determined on 10,000 bootstrapping iterations. In order to test whether faster tau accumulation rates were associated with faster cognitive decline in a memory composite (i.e., ADNI-MEM) available in ADNI[84], we computed annual change rates in cognition based using the same linear mixed model approach that was used for determining tau accumulation rates. Using linear regression controlling for age, sex and education, we then tested whether faster tau accumulation rates were associated with faster decline in ADNI-MEM. All analyses were computed using R statistical software (r-project.org[30]). For determining statistical significance, we applied an alpha threshold of 0.05 for all analyses. All regression weights are presented as standardized beta values which were determined using the lm.beta package in R (https://cran.r-project.org/web/packages/lm.beta/lm.beta.pdf).

## Reporting summary

Further information on research design is available in the Nature Research Reporting Summary linked to this article.

## Data availability

All data used in this manuscript are publicly available from the ADNI database (adni.loni.usc.edu) upon registration and compliance with the data use agreement. BioFINDER data are available from the principal investigator (O.H.), anonymized data will be shared by request from a qualified academic investigator for the sole purpose of replicating procedures and results presented in the article and as long as data transfer is in agreement with EU legislation on the general data protection regulation and decisions by the Ethical Review Board of Sweden and Region Skåne, which should be regulated in a material transfer agreement. The data that support the findings of this study are available on reasonable request from the corresponding author. Source data are provided with this paper.

## Code availability

An example version of the R code used for the main analysis can be found together with simulated data in the Supplementary (i.e., Supplementary Software 1).

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

## Acknowledgements

We acknowledge all members of the Alzheimer's Disease Neuroimaging Initiative. N.F. received research support from the Hertie Foundation (Network of Excellence in Clinical Neurosciences). O.H. has acquired research support (for the institution) from AVID Radiopharmaceuticals, Biogen, Eli Lilly, Eisai, GE Healthcare, Pfizer, and Roche. In the past 2 years, he has received consultancy/speaker fees from Roche, Genentech, Siemens, Biogen, Alzpath, and Cerveau. The BioFINDER study was supported by the Swedish Research Council, the Knut and Alice Wallenberg foundation, the Marianne and Marcus Wallenberg foundation, the Strategic Research Area MultiPark (Multidisciplinary Research in Parkinson's disease) at Lund University, the Swedish Alzheimer Foundation, the Swedish Brain Foundation, the Parkinson Foundation of Sweden, the Parkinson Research Foundation, the Skåne University Hospital Foundation, and the Swedish federal government under the ALF agreement. Doses of ¹⁸F-flutemetamol injection were sponsored by GE Healthcare. The precursor of ¹⁸F-flortaucipir was provided by AVID Radiopharmaceuticals. ADNI data collection and sharing for this project was funded by the ADNI (National

Institutes of Health grant U01 AG024904) and DOD ADNI (Department of Defense award number W81XWH-12-2-0012). ADNI is funded by the National Institute on Aging, the National Institute of Biomedical Imaging and Bioengineering, and through contributions from the following: AbbVie, Alzheimer's Association, Alzheimer's Drug Discovery Foundation, Araclon Biotech, BioClinica Inc., Biogen, Bristol-Myers Squibb Company, CereSpir Inc., Cogstate, Eisai Inc., Elan Pharmaceuticals Inc., Eli Lilly and Company, EuroImmun, F. Hoffmann-La Roche Ltd. and its affiliated company Genentech Inc., Fujirebio, GE Healthcare, IXICO Ltd., Janssen Alzheimer Immunotherapy Research and Development LLC., Johnson & Johnson Pharmaceutical Research and Development LLC., Lumosity, Lundbeck, Merck & Co. Inc., Meso Scale Diagnostics LLC., NeuroRx Research, Neurotrack Technologies, Novartis Pharmaceuticals Corporation, Pfizer Inc., Piramal Imaging, Servier, Takeda Pharmaceutical Company, and Transition Therapeutics. The Canadian Institutes of Health Research is providing funds to support ADNI clinical sites in Canada. Private sector contributions are facilitated by the Foundation for the National Institutes of Health (www.fnih.org).

## Author contributions

L.F.: data analysis, study concept and design, drafting the manuscript; M.E., M.B., R.O., M.D., and O.H.: study concept and design, critical revision of the manuscript; D.B., P.H., A.S., A.D., S.R., A.R., and O.S.: data preprocessing, data analysis, critical revision of the manuscript, K.B., D.J., and A.P.B.: critical revision of the manuscript; R.S. and N.M.C.: data acquisition, critical revision of the manuscript; N.F.: data preprocessing, data analysis, study concept and design, drafting the manuscript.

## Funding

## Competing interests

The authors declare no competing interests.
