## [Peer Review File · Nature Communications]

Earlier Alzheimer's disease onset is associated with tau pathology in brain hub regions and facilitated tau spreadingREVIEWER COMMENTS

Reviewer #1 (Remarks to the Author):

This is a very innovative and relevant study, extremely clear and well written, well conducted. I have some comments, two of which I consider as major (7 and 10).

1. The use of the term bias is ambiguous (e.g. in “a bias of tau-PET signal towards...”); it should not be used in the abstract, and it could be used in the article (core text) but after having been clearly defined (note that when saying “a bias of tau pathology deposition from allocortical medial temporal lobe regions towards neocortical fronto-parietal association cortices” this sounds clear/OK as this specifies what you mean by “bias” and versus what).
2. It is also classically proposed that earlier AD symptom onset is associated with faster progression because of greater reserve to start with so that at a similar level of symptoms the disease is much more advanced, and then shows a faster progression (as a result of the fact that the pathology spread tends to accelerate as the disease progresses). It would be I think interesting to 1) mention this well-known interpretation/hypothesis and 2) assess it or control for it in the analyses.
3. “a stronger bias of tau pathology towards fronto-parietal hub regions that are highly relevant for cognition may thus drive earlier symptom manifestation in AD »: Yes, but note as well that tau is more extended anyway, and instead of being the specific topography (ie fronto-parietal network), it could simply be the fact that tau is more extended, involves a much larger part of the brain.
4. “In addition, tau pathology deposition in globally connected hub regions may further accelerate the progression and spreading of tau pathology itself, which is in turn a strong driver of cognitive decline”: same comment as before; actually all the rationale could apply to the tau being more extended instead of tau being located in hub regions. I understand that when you take the 10% highest tau region (epicenters) this might in part take the extent into account, but only in part (and probably specifically only for bias towards tau location in hubs not for tau propagation) as this doesn't prevent from faster tau propagation being related to the fact that tau was more extended. Same line: If “tau spreads from circumscribed epicenters to connected brain regions”, I agree that tau would tend to propagate faster if epicenters are hub regions, but this would also be the case if there are more epicenters right?
5. Consequently, I would suggest to dissociate those two factors; it would be meaningful to show that, at a similar level of tau extent (ie considering people who have the same brain volume portion with tau), those with tau in hubs are younger and showed faster tau accumulation and cognitive decline.
6. Please define abbreviations in the Figures.
7. Please give the results when age=65 is set to separate early versus late onset patients as this sounds clinically meaningful; and provide group comparison results in addition to regression (eg btw age and hub gradient). This could be in a table or supplementary only if limited in word counts. Only after writing this I realized that there were only very few early-onset AD so this is probably not feasible. Maybe it could be done by using a cut-off of 70 yrs instead... I think this is a problem though and needs also to be pointed before (not speaking about early-onset / late onset in the introduction/rationale because this is misleading. It would help saying that here what is really assessed is younger to older age at onset – but not early versus late onset given that most (percentage?) included patients would be classified as late-onset. Acknowledging this right from the beginning would solve most of this issue I think.
8. “Results remained significant when repeating the analyses using robust regression or when additionally controlling for global Aβ levels or ApoE4 status” : please give the stats.
9. It would help to more clearly specify right from the beginning that all analyses are performed separately in two different cohorts and why (ie that the biofinder is a replication sample). Saying that

“we employed two independent samples covering the entire AD spectrum, including 242 participants of the Alzheimer’s disease neuroimaging initiative (ADNI) and 57 subjects of the BioFINDER study” doesn’t say that all analyses were replicated independently in these two independent samples, and neither why.

10. That the hippocampus is excluded from the analyses is really a major caveat in this study. The authors should 1) better explain why they think their results are still valuable: let’s say that late-onset AD tend to have tau mainly in the hippocampus, and even tau propagation mainly only in this structure for those at earliest tau stages, how could data be interpreted? When taking the 10% regions with highest tau, what about those participants. This is a huge bias in the data as the main difference between both groups is the fact that the hippocampus is spared (in early-onset), or is the main site of tau (in late-onset), so I am not convinced that the data remains strong given this caveat; 2) given the importance of this caveat, it is not possible that it only appears in the discussion; it is not only a methodological details that could go in the method section, it is a crucial aspect of data processing and the readers should know when they read the results section and understand the way the authors analyzed the data. Please find ways to convince (do further analyses, show at least to the reviewers why this is still correct if it is, explain why you think this might not have influenced/biased the results etc).

11. The only possible way to convincingly demonstrate that this has no major impact on the results could be, in my sense, to replicate with atrophy from T1 (instead of tau from PET) as i) almost all the rationale is also true for atrophy (specificity of topography and propagation rate according to age at onset), and ii) it would be devoid of issues related with partial volume effects and off-target sparing and would allow to keep the hippocampus included.

12. More details on lack of results in asymptomatic individuals would be interesting: do the processing method really apply to those participants (for instance step E in Fig 1 still OK in participants with only very small amount of tau?), what if only including those (participants) with tau, etc. Is the point being asymptomatic or the lack of tau; if there is a way to know I think it is important, and then it would seem important to rephrase accordingly (results/discussion related sections).

Same line: I was very surprised that “In patients with preclinical AD, we also found a significant association between a higher tau hub gradient at baseline and a faster subsequent tau accumulation rate in ADNI ($r=0.347$, $p=0.006$, Fig.4D, left panel)”; this suggests that all negative results in previous analyses were not due to the lack of tau in these patients.

13. “patients with younger symptomatic onset of AD show a stronger pattern of tau pathology deposition from inferior temporal non-hub regions towards globally connected hubs in the fronto-parietal association cortex (Fig.1D, Fig.2).”; it seems to me that it is misleading to cite fig 1D here; this figure illustrates the hubs not the results cited in this sentence. Please cite fig 2 only or cite fig1D differently/rephrase.

14. “suggesting that the spatial pattern rather than the mere extent of tau pathology may determine the likelihood, type and aggressiveness of symptom manifestation in AD” is it really something that we could say from the results? Please justify why the results show that it is the pattern rather than the mere extent; as far as I understand (and in link with my comments above) it could be both ie results are biased by the extent.

15. I am also wondering how much the presence of abeta (both locally, and or in connected brain regions) might have participated to faster tau propagation. In other words, the fact that tau propagates faster when initially present in hubs region could also be due to the fact that abeta is also more present in these hub regions (or in brain regions connected to those hub regions) (compared to non-hub regions).

16. “In a series of resting-state and task-fMRI studies in AD patients, we could previously show that the functional integrity of fronto-parietal control network hubs is critically important for maintaining cognitive function in AD.24, 25, 26, 27, 28, 29 “ and “which we have previously shown to be central for

maintaining cognitive performance in AD.24, 25, 26, 29" : not sure it is needed to cite so many references for the same sentence (and even later when specifying for each imaging modality); I would advise to cite one or two, and possible 1-2 from another group, sounds more convincing to cite less and useless to cite 4-6 but all from the same lab (especially for a same statement).

17. I had the feeling that the discussion was too long and could be more concise and efficient. There are repetitions, things that are said several times in different ways. I am sure that it could be shorten a little bit for improved efficiency for the reader.

18. In the limitation please add that tau-PET data were not corrected for partial volume effects.

Reviewer #2 (Remarks to the Author):

This manuscript by Frontzkowski et al. retrospectively analyzes resting-state fMRI and longitudinal tau PET data from separate cohorts to analyze links between tau deposition and fMRI-derived functional networks in the brain.

This is not the first study linking fMRI with longitudinal tau PET. In fact, another paper recently published in Nature Communications by Drs. Hansson and Franzmeier, authors on this manuscript, showed that resting state functional connectivity is associated with longitudinal tau covariance. The main contribution of this paper is highlighting the role that fMRI derived hubs play in younger symptomatic adults. The authors have created a hubness-weighted tau positivity metric and examined its association with age of symptom onset and annual tau accumulation rate. The weighting mechanism is guided by the intuition that a hub ROI could have a lower tau load and yet be a key contributor to disease spread. The new results have clinical significance. But there are several major concerns about the methodology and the interpretation of the results as outlined below:

(1) The casual use of the word "bias" everywhere in this manuscript makes me uncomfortable. As I understand, the authors are using the word bias to underscore the importance of hubs in the tau distribution. This has nothing to do with *statistical bias*. The metric that is actually being used is actually a measure of tau accumulation weighted by a hubness metric (node degree). I therefore strongly recommend rewording all parts of the manuscript, including the title, where the word "bias" is used casually and replacing it by "preferential accumulation" or using a more meaningful descriptive phrase like "regions simultaneously having high tau and high node degree/hubness."

(2) The methodology seems ad hoc without proper justifications and also very poorly described. Several specific concerns are described below.

a. From the description, it appears that in Fig. 1C, row averages were computed from the fMRI adjacency matrix. This is a mean connectivity measure for each ROI. What is being computed is "node degree" for a non-binary adjacency matrix. The non-technical descriptor "global functional connectivity" for node degree seems inappropriate and confusing.

b. The so-called "hub gradient" is not an accepted term in fMRI literature. Node degree is an accepted measure of "hubness." But there is no justification for scaling the node degree to a [-1,+1] range. In their previous Nature Comm paper, the authors had used a Fisher z-transformation of the correlation coefficient, which is a well-known statistic. However, the seemingly ad hoc hub gradient measure in the current manuscript needs more justification. What does a -1 value or a +1 value mean for this measure? This is not a mathematical "gradient." The nomenclature needs to be reconsidered. Since it is eventually being used as a weight, I would recommend renaming it as "hubness weight" or something similar. Are the scaling factors same for all ROIs or different for different ROIs? A mathematical equation should be provided in both Fig. 1 and the Methods section to ensure reproducibility of the results.

c. The term "tau gradient" seems to be a misnomer as well. It is a tau measure weighted by a node degree measure and then scaled. This is neither a spatial gradient and nor a longitudinal tau change measure. This is a ratio. It should be renamed. The denominator is termed "mean tau" – is this a mean over the subjects, the ROIs, or both? A mathematical equation should be provided in both Fig.

1 and the Methods section to ensure reproducibility of the results.

(3) It is perplexing that the hubness-weighted tau measure is not able to discriminate between different diagnostic groups (Table 1). This mandates a lengthier discussion since tau in certain ROIs have high discriminative power.

(4) β is the Type II error rate in binary hypothesis testing and cannot be negative. What do the reported negative β 's in page 9 mean for regression? If β is used to denote the regression coefficient, then it might be a good idea to use a different notation and also to report the α and β used to calculate the p-values.

(5) Figs. 2 and 3 report associations between a hubness-weighted tau measure for each subject vs. age. It should be noted that the weighting factors are the same for each subject and the connectivity measures are group-level. These are not individual connectivity measures. So technically what is being used is a preset linear combination of tau values to create a weighted tau measure for each subject. Since the metric is based on entry-wise multiplication, a node with a stronger hubness measure and low tau could have the same contribution to this metric as a node with weaker hubness and high tau. So, if anything, this metric tells me that hub nodes can get away with having less tau load and contributing to symptomatic disease, which makes sense. I am not sure why this is being repeatedly referred to as "tau PET bias" to the hubs.

(6) I am curious whether other linear combinations of the tau measures are possible that could have similar effects. Once again, if these were individual connectivity measures and the weights were different for each subject, my interpretation would be different. What we are looking at is the same linear weighting scheme applied to all ROIs of each subject to create a composite tau measure. Some reference weighting schemes need to be tested out for meaningful interpretation of the results.

(7) Hubness measures themselves are likely to have a strong age association. The results shown do not allow decoupling the fMRI from the tau to determine whether a similar trend could be revealed by hubness alone. This is a major concern. It would be helpful to perform this analysis in a subset of the ADNI data for which both tau PET and fMRI are available. That way the hubness effects and tau PET effects can be decoupled from each other and separately analyzed at an individual level. Such evidence would be more conclusive than what is provided.

(8) The tau epicenters for analysis need to be named. The epicenters are currently described only as "10% of brain regions with highest baseline tau-PET positivity."

(9) The results in Fig 4 again are separate for preclinical vs. symptomatic. The global tau change for symptomatic cases is higher than that for asymptomatic preclinical cases. So I would not be surprised to see similar associations for all baseline tau PET measures, not just hubness-weighted ones. For this to be convincing, appropriate references need to be reported. Also, this phrase needs to be justified: "tau epicenters fall in a hub region with many connections to other regions." This analysis is not done for the specific epicenters or specific fMRI hubs. This is for all ROIs and is capturing a mixed effect of tau and hubness. So, I am not sure we are seeing effects specific to individual tau epicenters or fMRI hubs in these results.

(10) While ADNI data can be accessed through their website, the terms for BioFINDER data sharing are very restrictive and would require contacting Dr. Hansson personally. To ensure reproducibility, it is recommended that scripts and data from both cohorts be made more easily accessible.

In summary, the idea that hub regions can have a strong impact on tau spread even when the tau load in these regions is small is intuitive and meaningful. But the interpretations are not necessarily saying what the plots are showing. Phrases like "stronger shift of tau pathology towards globally connected hub regions" and "stronger bias of tau pathology towards more globally connected hubs" need to be removed from this manuscript. Results based on population-level hubness weights need to be compared with alternative weighting schemes for benchmarking. Finally, evidence from subjects with both fMRI and tau PET available would greatly increase the value of this study.

REVIEWER COMMENTS

Reviewer #1 (Remarks to the Author):

This is a very innovative and relevant study, extremely clear and well written, well conducted. I have some comments, two of which I consider as major (7 and 10).

We thank the reviewer for these encouraging remarks

1. The use of the term bias is ambiguous (e.g. in “a bias of tau-PET signal towards...”); it should not be used in the abstract, and it could be used in the article (core text) but after having been clearly defined (note that when saying “a bias of tau pathology deposition from allocortical medial temporal lobe regions towards neocortical fronto-parietal association cortices” this sounds clear/OK as this specifies what you mean by “bias” and versus what).

Response: *We agree with the reviewer that the term “bias” can be ambiguous and misleading if not explained in further detail. Please also note that Reviewer 2 (comment 1) has raised the same concern. Thus, we have changed our terminology to “tau hub ratio”, throughout the manuscript since the measure that we determined is in fact a weighted ratio of tau in hubs vs. non-hubs that is adjusted for overall tau-PET levels. Further, we have simplified our wording in the abstract and title, and simply refer to “stronger tau pathology in hub regions”.*

2. It is also classically proposed that earlier AD symptom onset is associated with faster progression because of greater reserve to start with so that at a similar level of symptoms the disease is much more advanced, and then shows a faster progression (as a result of the fact that the pathology spread tends to accelerate as the disease progresses). It would be I think interesting to 1) mention this well-known interpretation/hypothesis and 2) assess it or control for it in the analyses.

Response: *This is an interesting thought and we are aware that inter-individual differences in reserve/resilience may account for a certain variability in symptom onset and disease trajectories. To account for potential confounding factors of reserve/resilience, all analyses presented in the current manuscript were controlled for years of education, which is a well-established proxy of reserve/resilience and which was consistently available for both the ADNI and BioFINDER sample. Further, we did not find an association between years of education and the tau hub ratio in A β + subjects (ADNI: β =-0.138, p =0.11; BioFINDER: β =0.166, p =0.286, linear regression, controlling for age, sex and diagnosis), suggesting that AD patients with high/low Education do not differ on the tau hub ratio. For the current study, we refrained from specifically testing modulating effects of reserve/resilience factors (e.g. education) on the association between tau pathology and symptom manifestation or symptom, since this would be a dedicated own research question and further complicate an already complex set of analyses. However, we now mention the possibility of potential reserve/resilience effects which may additionally influence symptom onset and clinical progression in the discussion section (p.17).*

3. “a stronger bias of tau pathology towards fronto-parietal hub regions that are highly relevant for cognition may thus drive earlier symptom manifestation in AD »: Yes, but note as well that tau is more extended anyway, and instead of being the specific topography (ie fronto-parietal network), it could simply be the fact that tau is more extended, involves a much larger part of the brain.

Response: *This is an important comment, and we agree that more widespread tau pathology rather than tau being more pronounced in a particular brain network may also lead to faster tau accumulation. We are aware of this potential confound, hence we have adjusted our tau hub ratio measure for global tau-PET levels (see Fig.1L). Supporting the view that the tau hub ratio does not simply reflect more widespread tau pathology but rather the spatial distribution pattern of tau-PET, we did not find that the tau hub ratio increases across disease severity (i.e. p .8: “Using an ANOVA, no difference in the tau hub gradient was found between diagnostic groups (table 1, ADNI, p =0.090, BioFINDER, p =0.566), suggesting that there is no systematic increase in the tau hub ratio across increasing diagnostic severity.”). Nevertheless, we agree with the reviewer that further analyses should be performed to clarify whether our results are influenced by overall higher tau levels in younger patients with symptomatic AD. For a description of these additional analyses, please see our response on comment 5.*

4. “In addition, tau pathology deposition in globally connected hub regions may further accelerate the progression and spreading of tau pathology itself, which is in turn a strong driver of cognitive decline”: same comment as before; actually all the rationale could apply to the tau being more extended instead of tau being located in hub regions. I understand that when you take the 10% highest tau region (epicenters) this might in part take the extent into account, but only in part (and probably specifically only for bias towards tau location in hubs not for tau propagation) as this doesn't prevent from faster tau propagation being related to the fact that tau was more extended. Same line: If “tau spreads from circumscribed epicenters to connected brain regions”, I agree that tau would tend to propagate faster if epicenters are hub regions, but this would also be the case if there are more epicenters right?

Response: *This is an excellent point, and we agree that at a given level of overall tau pathology, patients with more globally connected tau epicenters should show faster subsequent tau accumulation. Thus, we have performed a sliding window analysis from low to global high tau levels (i.e. to match the levels of tau pathology constant within a given window), and tested whether more globally connected tau epicenters are associated with faster tau*

accumulation within each window (i.e. at a given level of baseline tau pathology). Please see our response to the next comment for clarification of this analysis.

5. Consequently, I would suggest to dissociate those two factors; it would be meaningful to show that, at a similar level of tau extent (i.e. considering people who have the same brain volume portion with tau), those with tau in hubs are younger and showed faster tau accumulation and cognitive decline.

Response: This is an excellent suggestion. To address the reviewers' comment, we performed a sliding window analyses from low to high global tau levels in $A\beta+$ subjects, where we compared annual tau accumulation rates within each window for patients with a high (>median) vs. low (<median) tau hub ratio. Specifically, we sorted our sample from low to high global tau-PET levels (i.e. the average tau-PET positivity across the entire cortex). We then used a window size of 30% of the entire $A\beta+$ sample, which was shifted in steps of 3 subjects from low to high global tau-PET levels. Within each window, we quantified the mean annual global tau-PET change for subjects with a high (i.e. >median) tau hub ratio and in subjects with a low (i.e. <median) tau hub ratio. With this analysis approach, we were able to test whether at a given level of baseline tau levels, a higher tau hub ratio was associated with faster subsequent tau accumulation, i.e. to specifically determine whether the topographical distribution and not the overall severity of tau in hubs is associated with accelerated tau accumulation. To statistically test this hypothesis, we used linear regression to assess the interaction between baseline global tau-PET positivity (i.e. determined within each sliding window sample) and the tau hub ratio on annual global tau-PET increases. Here, we found a significant interaction, where there was faster tau-PET increase at a given level of baseline global tau-PET in patients with a high tau hub ratio compared to patients with a low tau hub ratio (ADNI: $\beta=-2.25$, $p=0.002$; BioFINDER: $\beta=-1.12$, $p=0.028$, Supplementary Figure 2A&B). In a similar vein, we tested whether at a similar level of baseline global tau-PET, patients with a high tau hub ratio were younger than patients with a low tau hub ratio. Supporting this view, we found a significant interaction between baseline global tau-PET and the tau hub ratio on age. In addition, we used this sliding-window approach to test whether at a given level of baseline tau pathology, symptomatic AD patients with a higher tau hub ratio were younger. Supporting this, we also found an interaction between baseline tau-PET and the tau hub ratio on age, where patients with higher baseline tau levels and a higher tau hub ratio were younger compared to individuals with the same overall tau load but a lower tau hub ratio (ADNI: $\beta=3.87$, $p=0.016$; BioFINDER: $\beta=0.43$, $p=0.038$, Supplementary Figure 2C&D). Further, we assessed whether at a given level of tau pathology, a higher tau hub ratio was associated with accelerated cognitive decline. This analysis was restricted to ADNI, where longitudinal cognitive data were available. Here, we found a significant interaction between baseline tau-PET and the tau hub ratio on annual change rates in ADNI-MEM ($\beta=3.29$, $p<0.001$, Supplementary Figure 2G), supporting the view that at a given level of tau pathology, a higher tau hub ratio is associated with faster cognitive changes. Lastly, we used the same analytical framework to test whether more globally connected tau epicenters are associated with faster tau accumulation at a given level of global tau levels (see previous comment). Here, we found a significant interaction between baseline tau-PET and tau epicenter connectivity on subsequent tau accumulation, where at a given level of baseline tau-PET, more globally interconnected epicenters were associated with faster subsequent tau accumulation (ADNI: $\beta=2.54$, $p=0.001$; BioFINDER: $\beta=0.33$, $p<0.001$, Supplementary Figure 2E&F). We believe that these additional results strengthen the view that the topological distribution of tau pathology in hubs may facilitate earlier symptom manifestation, faster tau accumulation and faster cognitive decline. We have added these novel analyses to the statistics (p.27), results (p.12/13) and supplementary section (i.e. supplementary Figure 2) of the manuscript.

6. Please define abbreviations in the Figures.

Response: All abbreviations are now described and defined in the figure legends.

7. Please give the results when age=65 is set to separate early versus late onset patients as this sounds clinically meaningful; and provide group comparison results in addition to regression (eg btw age and hub gradient). This could be in a table or supplementary only if limited in word counts. Only after writing this I realized that there were only very few early-onset AD so this is probably not feasible. Maybe it could be done by using a cut-off of 70 yrs instead... I think this is a problem though and needs also to be pointed before (not speaking about early-onset / late onset in the introduction/rationale because this is misleading. It would help saying that here what is really assessed is younger to older age at onset – but not early versus late onset given that most (percentage?) included patients would be classified as late-onset. Acknowledging this right from the beginning would solve most of this issue I think.

Response: Following the reviewers' suggestion, we have first of all removed changed our terminology and avoid the term "early-onset" as this is typically related to a symptom onset before 65 years. In addition, we ran additional analyses, comparing global tau accumulation, the tau hub ratio, and epicenter connectivity at an age cut-off of 70 years. In ADNI, there were a total of 16 symptomatic AD patients below 70 years and 73 symptomatic AD patients above 70 years of age. In BioFINDER, there were 8 symptomatic AD patients below 70 and 17 above 70. In line with our main analyses using age as a continuous measure, we find that patients below 70 show a higher tau hub ratio (ADNI: $p=0.097$, BioFINDER: $p=0.001$), more globally connected tau epicenters (ADNI: $p=0.002$, BioFINDER: $p<0.001$) and a faster tau accumulation rate (ADNI: $p=0.098$, BioFINDER: $p<0.001$). All group differences were computed using ANCOVAs controlling for sex, diagnosis and education. Results were stronger in BioFINDER than in ADNI. Yet, we caution that these analyses are underpowered due to a highly imbalanced sample size between the below and above 70 group. Since the linear model approach using age as a continuum does not suffer from

this imbalance, we kept the linear model approach for the main manuscript and retain this exploratory age split at 70 years in the rebuttal, which will be made available as a supplementary file in case the manuscript is accepted for publication. We have added a sentence to the discussion section, stating specifically that our studies' conclusions are mostly limited to the age range of typical late onset AD patients (p. 17).

8. "Results remained significant when repeating the analyses using robust regression or when additionally controlling for global A β levels or ApoE4 status" : please give the stats.

Response: We have included the statistics for regression models including global A β levels or ApoE4 in supplementary table 1.

9. It would help to more clearly specify right from the beginning that all analyses are performed separately in two different cohorts and why (ie that the biofinder is a replication sample). Saying that "we employed two independent samples covering the entire AD spectrum, including 242 participants of the Alzheimer's disease neuroimaging initiative (ADNI) and 57 subjects of the BioFINDER study" doesn't say that all analyses were replicated independently in these two independent samples, and neither why.

Response: Thanks for this suggestion. We have now made clear in the abstract and introduction that all results were independently replicated across both samples.

10. That the hippocampus is excluded from the analyses is really a major caveat in this study. The authors should 1) better explain why they think their results are still valuable: let's say that late-onset AD tend to have tau mainly in the hippocampus, and even tau propagation mainly only in this structure for those at earliest tau stages, how could data be interpreted? When taking the 10% regions with highest tau, what about those participants. This is a huge bias in the data as the main difference between both groups is the fact that the hippocampus is spared (in early-onset), or is the main site of tau (in late-onset), so I am not convinced that the data remains strong given this caveat; 2) given the importance of this caveat, it is not possible that it only appears in the discussion; it is not only a methodological details that could go in the method section, it is a crucial aspect of data processing and the readers should know when they read the results section and understand the way the authors analyzed the data. Please find ways to convince (do further analyses, show at least to the reviewers why this is still correct if it is, explain why you think this might not have influenced/biased the results etc).

Response: The reviewer is correct that the hippocampus has been excluded from our initial analyses due to off-target binding of the AV1451 tau-PET tracer to the choroid plexus, which is spatially adjacent to the hippocampus and can therefore confound the hippocampal tau-PET signal.¹ Further, the reviewer is correct that the hippocampus is an early site of tau pathology (i.e. Braak II), where significant tau accumulation occurs typically subsequent to entorhinal tau deposition (i.e. Braak I)² and prior to neocortical tau deposition, e.g. in the inferior temporal lobe (i.e. Braak III) and beyond (i.e. Braak IV-VI), as also confirmed by a recent study using the RO948 2nd generation tau-PET tracer which is less affected by off-target binding in the medial temporal lobe.³ Since the entorhinal cortex is considered a key brain region from which tau pathology spreads to the hippocampus across anterograde connections,² tau levels in the entorhinal cortex and hippocampus are highly associated in patients with abnormal brain amyloid deposition.⁴ Further higher entorhinal tau levels predict lower hippocampal synaptic density⁵ and lower hippocampal volume,^{3,6} suggesting that tau-associated pathological brain changes are highly similar between the entorhinal cortex and hippocampus. Entorhinal tau is assumed to precede hippocampal tau, and hippocampal tau is assumed to precede inferior temporal tau deposition (e.g. parahippocampal cortex), reflecting the gradual spread of tau pathology across connected brain regions. Importantly, both the entorhinal and parahippocampal cortex are included in our 200 ROI brain parcellation, which should therefore serve as an indirect proxy for hippocampal tau pathology, since tau hippocampal tau levels are correlated with tau levels in the remaining temporal lobe. Thus, differences in medial temporal lobe tau should be also reflected in differences in entorhinal and parahippocampal tau differences and therefore influence the tau hub ratio. Therefore, we conclude that our results should not be highly biased by the missing hippocampal tau-PET signal. Nevertheless, we agree with the reviewer that investigating the role and impact of hippocampal tau levels on our analyses is interesting and requires further investigation. To this end, we have decided to employ a targeted approach which allows to separate the hippocampal tau signal from off target binding regions, which we have established previously for the Flortaucipir tau-PET tracer.¹ For details on these analyses, please see our response on the next comment.

11. The only possible way to convincingly demonstrate that this has no major impact on the results could be, in my sense, to replicate with atrophy from T1 (instead of tau from PET) as i) almost all the rationale is also true for atrophy (specificity of topography and propagation rate according to age at onset), and ii) it would be devoid of issues related with partial volume effects and off-target sparing and would allow to keep the hippocampus included.

Response: To include the hippocampus in our analyses, the reviewer suggests replicating all analyses using longitudinal T1-weighted MRI data as a proxy of tau pathology. However, we believe that this suggestion has several shortcomings. First, we agree that tau is a key driver of neurodegeneration which manifests as grey matter atrophy on T1 MRI as suggested by previous studies, showing that tau deposition patterns obtained via tau-PET predict preceding or subsequent patterns of grey matter atrophy.^{7,8} However, these studies used tau-PET as a specific

marker for AD-related tau pathology as a predictor of future or preceding neurodegeneration, but did not use baseline atrophy maps as a predictor of future atrophy. Thus, tau-PET would still be needed at baseline to forecast future atrophy patterns, hence including the hippocampus would still be complicated by off-target binding of the tau-PET tracer.

Yet, this poses another problem, since atrophy lags temporally behind the spreading of tau pathology,⁹ i.e. tau spreads first, while atrophy follows at a later timepoint once sufficient tau pathology has accumulated.⁹ Thus, atrophy would most likely be observed in those regions in which significant tau pathology is already observed at baseline (e.g. hubs) rather than in those brain regions connected to the sites of highest baseline tau-PET.

Second, grey matter atrophy is not specific in AD and can be exacerbated by multiple co-pathologies including vascular disease or other neurodegenerative proteinopathies (e.g. Lewy Bodies, TDP-43 etc.), which would make a specific link to tau pathology difficult. In addition, our sample size would drastically drop since only a subset of the participants included in the current study have available longitudinal T1w MRI data. Thus, we prefer to keep our main analyses using tau-PET data.

Nevertheless, we tried to address the reviewers concern using an alternative approach that allows to quantify Flortaucipir tau-PET in the hippocampus while minimizing the off-target binding effects introduced by the adjacent choroid plexus. Specifically, we have shown previously that excluding a pre-defined choroid plexus mask from the hippocampus can help correct the hippocampal tau-PET signal for off-target binding and lead to improved diagnostic accuracy (i.e. discrimination between cognitively unimpaired and cognitively impaired individuals) and stronger correlations between hippocampal tau-PET signal and cognitive scores.¹ Thus, we have additionally extracted hippocampal tau-PET SUVRs using a hippocampal ROI from which the choroid plexus mask was excluded to minimize choroid plexus binding. This was performed on unsmoothed tau-PET images to further minimize signal spill over. Including two hippocampal ROIs increased the total number of ROIs in our brain parcellation to 202, hence we further recomputed functional connectivity for this 202 ROI parcellation using the 1000 HCP participants. Subsequently, we adopted the same analysis pipeline illustrated in Figure 1, in order to determine a tau hub ratio, this time also including the hippocampal ROIs. Please note also that our Gaussian Mixture Modelling approach to clean on-target from off-target binding helps further clean remaining unspecific signal from the hippocampal tau-PET signal. Using these new data, we recomputed all analyses included in the initially submitted version of the manuscript. As expected, all results remained fully consistent with our main analyses including a higher tau hub ratio in younger symptomatic AD patients, more globally connected tau epicenters in younger AD patients as well as faster annual tau accumulation in patients with a higher tau hub ratio. We have included these additional analyses in the methods (p.22), results (p.8-13) and supplementary (i.e. supplementary table 2) of the revised manuscript, and briefly mention them in the discussion (p.17).

12. More details on lack of results in asymptomatic individuals would be interesting: do the processing method really apply to those participants (for instance step E in Fig 1 still OK in participants with only very small amount of tau?), what if only including those (participants) with tau, etc. Is the point being asymptomatic or the lack of tau; if there is a way to know I think it is important, and then it would seem important to rephrase accordingly (results/discussion related sections).

Same line: I was very surprised that “In patients with preclinical AD, we also found a significant association between a higher tau hub gradient at baseline and a faster subsequent tau accumulation rate in ADNI ($\beta=0.347$, $p=0.006$, Fig.4D, left panel)”; this suggests that all negative results in previous analyses were not due to the lack of tau in these patients.

Response: This comment relates to the fact that we did not find consistent associations between the tau hub ratio and age or tau accumulation in individuals with preclinical AD. First, the Gaussian mixture modeling approach to clean tau-PET SUVRs from off-target signal is well suited to also capture relatively low but meaningful tau levels, since the fitting is performed across the entire sample of amyloid negative and amyloid positive individuals per ROI. To illustrate this, we can compare the tau-PET levels in the entorhinal cortex (i.e. the typical site where earliest tau pathology is found in AD) between cognitively normal $A\beta^-$ subjects (i.e. healthy controls) vs. cognitively normal $A\beta^+$ subjects (i.e. preclinical AD). Using regular tau-PET SUVRs, there is a Cohens d difference of 0.48 in entorhinal tau-PET between controls and preclinical AD, while this effect size increases to 0.63 using tau-PET positivity in BioFINDER. In ADNI, the Cohen's d increases from 0.52 to 0.62 when using tau positivity instead of tau-PET SUVRs. This suggests that our gaussian mixture model approach may in fact help to identify early-stage tau abnormalities and does not eliminate low tau-PET signal.

Potentially, the inconsistency in the association between the tau hub ratio and annual tau accumulation in preclinical AD between ADNI and BioFINDER is driven by the fact that ADNI controls have slightly higher tau-PET levels than BioFINDER (see table 1). For instance, ADNI CN $A\beta^+$ subjects have a mean tau-PET SUVR of 1.15 ± 0.1 in a temporal lobe meta ROI that captures early tau, whereas BioFINDER CN $A\beta^+$ subjects have only a temporal tau-PET SUVR of 1.09 ± 0.05 . Thus, preclinical AD patients in ADNI are slightly more advanced in terms of tau progression than preclinical AD patients in BioFINDER. A recent study has found that tau accumulation accelerates drastically once tau has spread from the entorhinal cortex to the inferior temporal gyrus.¹⁰ Since preclinical AD participants in ADNI participants show slightly higher temporal lobe tau levels they may be closer to the tau acceleration phase, which is supported by the fact that ADNI participants show at least some annual tau accumulation (see the range of annual tau change rates on the y-axis in Fig4, panel D, left), while there is almost no annual tau accumulation in preclinical AD patients in BioFINDER (Figure 4, panel E, left).

This suggests that a certain level of tau pathology at baseline is needed, before significant tau accumulation can be observed longitudinally. We have added a sentence to the discussion section, clarifying that preclinical AD

patients in BioFINDER have less tau than in ADNI and may therefore not show significant tau accumulation during the follow-up observational period (p.16 bottom-17).

13. "patients with younger symptomatic onset of AD show a stronger pattern of tau pathology deposition from inferior temporal non-hub regions towards globally connected hubs in the fronto-parietal association cortex (Fig.1D, Fig.2)."; it seems to me that it is misleading to cite fig 1D here; this figure illustrates the hubs not the results cited in this sentence. Please cite fig 2 only or cite fig1D differently/rephrase.

Response: We have removed Fig.1D from the sentence, thanks for picking this up.

14. "suggesting that the spatial pattern rather than the mere extent of tau pathology may determine the likelihood, type and aggressiveness of symptom manifestation in AD" is it really something that we could say from the results? Please justify why the results show that it is the pattern rather than the mere extent; as far as I understand (and in link with my comments above) it could be both ie results are biased by the extent.

Response: This is an important comment. In order to assess whether the pattern rather than the mere extent of tau pathology is associated with the subsequent rate of tau spreading, we have performed sliding window analyses across the spectrum of global tau-PET levels, i.e. we defined equally sized groups from low to high tau-PET and split within each group regarding a high vs. low tau hub ratio. Here, we could show that at a given level of global tau pathology, patients with a higher tau hub ratio show faster tau accumulation. This suggests that the pattern of tau deposition indeed influences the rate of tau accumulation, where subjects with stronger tau in hub regions show faster tau accumulation. Please see our response on comment 5, where we clarify this in detail.

15. I am also wondering how much the presence of abeta (both locally, and or in connected brain regions) might have participated to faster tau propagation. In other words, the fact that tau propagates faster when initially present in hubs region could also be due to the fact that abeta is also more present in these hub regions (or in brain regions connected to those hub regions) (compared to non-hub regions).

Response: This is an interesting comment and we fully agree that investigating the role of Abeta in tau spreading is key to understand how both pathologies interact, but we would also like to highlight that this is a research question on its own that goes beyond the scope of the current study. Please note that we specifically investigate the role of Abeta in tau spreading in an ongoing dedicated research project in our lab in Munich, which will be summarized in a dedicated manuscript once all analyses are concluded. To address the reviewers' comment, however, we assessed the same hub ratio also for amyloid-PET data, to determine whether amyloid is located preferentially in hubs or non-hubs in a given AD patient and whether amyloid in hubs is associated with the rate of tau accumulation and age in symptomatic AD. Here, we did not find that a higher amyloid hub ratio predicted faster subsequent tau accumulation, neither in preclinical (ADNI: $\beta=-0.010$, $p=0.943$; BioFINDER: $\beta=-0.248$, $p=0.406$) nor in clinical AD (ADNI: $\beta=0.086$, $p=0.432$; BioFINDER: $\beta=0.018$, $p=0.937$), using linear regression controlling for age and sex. Also, we did not find an association between younger age and a higher amyloid hub ratio, neither in preclinical (ADNI: $\beta=0.139$, $p=0.295$; BioFINDER: $\beta=0.258$, $p=0.445$) nor in clinical AD (ADNI: $\beta=-0.09$, $p=0.409$; BioFINDER: $\beta=0.046$, $p=0.844$), suggesting that younger age is not associated with stronger amyloid pathology in hubs in symptomatic AD patients. When recomputing our main analyses (i.e. the effect of tau in hubs at baseline as a predictor of future tau accumulation) while controlling for the amyloid hub ratio, we still find a significant effect of the tau hub ratio on future tau accumulation in patients with clinical AD (ADNI: $\beta=0.319$, $p=0.009$; BioFINDER: $\beta=0.738$, $p=0.001$). These analyses support our main conclusion that faster tau accumulation and younger age in symptomatic AD is associated with stronger tau in hubs. Since these analyses on the role of amyloid in tau spreading are beyond the scope of the current study, we did not include them in the main manuscript. However, these results will be made available online in the rebuttal in case the manuscript is accepted for publication.

16. "In a series of resting-state and task-fMRI studies in AD patients, we could previously show that the functional integrity of fronto-parietal control network hubs is critically important for maintaining cognitive function in AD.24, 25, 26, 27, 28, 29 " and "which we have previously shown to be central for maintaining cognitive performance in AD.24, 25, 26, 29" : not sure it is needed to cite so many references for the same sentence (and even later when specifying for each imaging modality); I would advise to cite one or two, and possible 1-2 from another group, sounds more convincing to cite less and useless to cite 4-6 but all from the same lab (especially for a same statement).

Response: We thank the reviewer for this comment and have removed several of our own publications, while adding papers from other groups. We now state that "In a series of resting-state fMRI studies, we and others could previously show that the functional integrity of fronto-parietal control network hubs is critically important for maintaining cognitive function in aging,¹¹ AD¹²⁻¹⁴ and other neurodegenerative diseases.¹⁵

17. I had the feeling that the discussion was too long and could be more concise and efficient. There are repetitions, things that are said several times in different ways. I am sure that it could be shorten a little bit for improved efficiency for the reader.

Response: *We have tried to shorten the discussion where possible.*

18. In the limitation please add that tau-PET data were not corrected for partial volume effects.

Response: *We have added this as a limitation to the discussion section, while highlighting studies which showed that longitudinal tau-PET changes can also be detected without partial volume correction (p. 18).*

Reviewer #2 (Remarks to the Author):

This manuscript by Frontzkowski et al. retrospectively analyzes resting-state fMRI and longitudinal tau PET data from separate cohorts to analyze links between tau deposition and fMRI-derived functional networks in the brain.

This is not the first study linking fMRI with longitudinal tau PET. In fact, another paper recently published in Nature Communications by Drs. Hansson and Franzmeier, authors on this manuscript, showed that resting state functional connectivity is associated with longitudinal tau covariance. The main contribution of this paper is highlighting the role that fMRI derived hubs play in younger symptomatic adults. The authors have created a hubness-weighted tau positivity metric and examined its association with age of symptom onset and annual tau accumulation rate. The weighting mechanism is guided by the intuition that a hub ROI could have a lower tau load and yet be a key contributor to disease spread. The new results have clinical significance. But there are several major concerns about the methodology and the interpretation of the results as outlined below:

(1) The casual use of the word “bias” everywhere in this manuscript makes me uncomfortable. As I understand, the authors are using the word bias to underscore the importance of hubs in the tau distribution. This has nothing to do with *statistical bias*. The metric that is actually being used is actually a measure of tau accumulation weighted by a hubness metric (node degree). I therefore strongly recommend rewording all parts of the manuscript, including the title, where the word “bias” is used casually and replacing it by “preferential accumulation” or using a more meaningful descriptive phrase like “regions simultaneously having high tau and high node degree/hubness.”

Response: We thank the reviewer for this comment and would like to note that Reviewer 1 had a similar concern. Thus, we have modified our wording and do not make use of the word bias anymore in the revised version of the manuscript. Also, we have omitted the word “gradient” as we understand from both reviewers’ comments, that this terminology may not represent the data correctly. Instead, we now use “tau hub ratio” as the main term to describe our metric that determines whether subject-level tau deposition patterns are more representative of a “hub-like” or “non-hub-like” pattern.

(2) The methodology seems ad hoc without proper justifications and also very poorly described. Several specific concerns are described below.

a. From the description, it appears that in Fig. 1C, row averages were computed from the fMRI adjacency matrix. This is a mean connectivity measure for each ROI. What is being computed is “node degree” for a non-binary adjacency matrix. The non-technical descriptor “global functional connectivity” for node degree seems inappropriate and confusing.

Response: Following the reviewers’ comment, we have changed our wording to “weighted degree”. However, we would like to note that the term global connectivity has been introduced by Cole et al., in 2012,¹⁶ and has since then been frequently used by us in published manuscripts.^{12,13,17} Therefore, we also mention the term “global connectivity” but use the label weighted degree in the figures and the main manuscript when referring to the parameter directly.

b. The so-called “hub gradient” is not an accepted term in fMRI literature. Node degree is an accepted measure of “hubness.” But there is no justification for scaling the node degree to a [-1,+1] range. In their previous Nature Comm paper, the authors had used a Fisher z-transformation of the correlation coefficient, which is a well-known statistic. However, the seemingly ad hoc hub gradient measure in the current manuscript needs more justification. What does a -1 value or a +1 value mean for this measure? This is not a mathematical “gradient.” The nomenclature needs to be reconsidered. Since it is eventually being used as a weight, I would recommend renaming it as “hubness weight” or something similar. Are the scaling factors same for all ROIs or different for different ROIs? A mathematical equation should be provided in both Fig. 1 and the Methods section to ensure reproducibility of the results.

Response: Following the reviewers’ comment, we have changed our labeling to “weighted degree” instead of global connectivity (see comment above). We’re further happy to explain why we rescaled the hub vector between -1 and 1. This scaling was purely done in order to later weight the tau-PET measures so that high tau-PET values in non-hub-like regions are assigned a negative weight, while high tau-PET values in more hub-like regions are assigned a positive weight. Multiplying the tau values with the node degree rescaled between -1 and 1 and averaging the resulting vector allows to determine whether tau pathology is more pronounced in hubs (i.e. if the resulting mean is positive) or in non-hubs (i.e. if the resulting mean is negative). In essence, this approach determines a ratio of tau in hubs vs. non-hubs while taking into account the connectivity strength of the hub region rather than binarizing into hubs vs. non-hubs. Thus, we now use the term “tau hub ratio” instead of “tau gradient”. Please note further that this ratio is then adjusted to the global tau-PET for each individual, in order to reduce the likelihood that this metric simply reflects disease severity but rather the spatial deposition pattern of tau (see our response on comment 3, reviewer 2). We have clarified this approach in the results section of the manuscript (p.8). In addition, we have added an equation to Figure 1 (panel M) and the methods section.

c. The term “tau gradient” seems to be a misnomer as well. It is a tau measure weighted by a node degree measure and then scaled. This is neither a spatial gradient and nor a longitudinal tau change measure. This is a ratio. It should be renamed. The denominator is termed “mean tau” – is this a mean over the subjects, the ROIs, or both? A mathematical equation should be provided in both Fig. 1 and the Methods section to ensure reproducibility of the results.

Response: We have changed the term tau hub gradient to tau hub ratio, which is a better description of the measure that we were using. Mean tau refers to the mean tau burden for a given subject, since the parameter is determined for each subject individually based on the patients' tau-PET data and the connectivity data from the HCP dataset. Overall, the measure describes whether abnormal tau levels in a given individual are more strongly expressed in hubs or non-hubs, while adjusting for global tau. For clarification, we have added an equation to Figure 1, panel M and the methods section.

(3) It is perplexing that the hubness-weighted tau measure is not able to discriminate between different diagnostic groups (Table 1). This mandates a lengthier discussion since tau in certain ROIs have high discriminative power.

Response: This is an important comment. We fully agree that tau has high discriminative power, since global and temporal lobe tau-PET levels show a strong and significant increase across diagnostic groups (see table 1). However, we specifically designed the tau hub ratio measure so that it does not capture increasing global tau levels and therefore does not systematically increase across diagnostic groups. Specifically, we adjusted the tau hub ratio by the global tau levels of each individual in order to avoid that it reflects disease severity (see Figure 1L&M). The motivation for adjusting the tau hub ratio was, that we wanted the tau hub ratio to specifically reflect the spatial distribution of the tau-PET signal (i.e. in hubs vs. non hubs or balanced across both) rather than the overall severity of tau pathology. We have added a sentence to the results section for clarification.

(4) β is the Type II error rate in binary hypothesis testing and cannot be negative. What do the reported negative β 's in page 9 mean for regression? If β is used to denote the regression coefficient, then it might be a good idea to use a different notation and also to report the α and β used to calculate the p-values.

Response: Beta (β) is the common label for standardized regression coefficients (compared to the unstandardized regression coefficient B together with the standard error SE), which we and others have consistently used in previous studies.¹⁸ Standardized regression weights are determined in R using the *lm.beta* package (<https://cran.r-project.org/web/packages/lm.beta/lm.beta.pdf>). We prefer using standardized regression weights over unstandardized regression weights, since standardized regression weights are easier to interpret in terms of effect size and facilitate comparability across studies and different scales. For a good overview on the comparison between standardized and unstandardized regression coefficients please see the following summary: <https://www.sciencedirect.com/topics/mathematics/standardized-regression-coefficient> We have clarified the usage of β in the statistics section of the manuscript (p.28) as well as in the figure legends. Please note that a common alpha threshold of 0.05 was used as a cut-off for statistical significance to determine two-sided p-values. We have added this information to the statistics section of the manuscript (p.28).

(5) Figs. 2 and 3 report associations between a hubness-weighted tau measure for each subject vs. age. It should be noted that the weighting factors are the same for each subject and the connectivity measures are group-level. These are not individual connectivity measures. So technically what is being used is a preset linear combination of tau values to create a weighted tau measure for each subject. Since the metric is based on entry-wise multiplication, a node with a stronger hubness measure and low tau could have the same contribution to this metric as a node with weaker hubness and high tau. So, if anything, this metric tells me that hub nodes can get away with having less tau load and contributing to symptomatic disease, which makes sense. I am not sure why this is being repeatedly referred to as "tau PET bias" to the hubs.

Response: As pointed out in our comments above, we have removed the term bias from the manuscript and have relabeled the metric to tau hub ratio. In addition, we ran further analyses using individual connectivity measures in ADNI (see our response on comment 7), yielding consistent results with our main analyses using the same HCP-based weighting scheme for all individuals. Nevertheless, the reviewer is fully correct that one of the core findings is that a hub region can get away with less tau while still contributing to disease manifestation and progression.

(6) I am curious whether other linear combinations of the tau measures are possible that could have similar effects. Once again, if these were individual connectivity measures and the weights were different for each subject, my interpretation would be different. What we are looking at is the same linear weighting scheme applied to all ROIs of each subject to create a composite tau measure. Some reference weighting schemes need to be tested out for meaningful interpretation of the results.

Response: This is an important comment, and we agree that reference weighting schemes can be helpful to determine the robustness of our findings compared to other possible linear combinations of connectivity and tau measures. To this end, we have adapted an iterative shuffling approach that we have used previously for connectivity vs. tau-PET analyses.¹⁹ Specifically, we generated new tau hub ratios by combining subject-level tau-PET data with randomly shuffled connectivity values. With this approach, we can test whether it is the unique combination of the actual topology of brain hubs and individual tau-PET data that drives our results. Specifically, we took the rescaled tau hub vector ranging between -1 and 1 and randomly shuffled it to determine a null-model hub vector. For each subject, we then multiplied this randomly shuffled tau hub vector with the individual tau-PET data and created the same tau hub ratio that is illustrated in Figure 1. We then re-ran our regression models on the association between age and the tau hub ratio as well as between the tau hub ratio and the annual tau accumulation

rate and saved the respective beta values of the association. This procedure was repeated 1000 times, each time generating a new randomly shuffled tau hub vector and an according tau hub ratio. We then performed an exact test to determine whether the actual beta values of the association between age and the true tau hub ratio were lower or higher than those 1000 beta values derived from the models using the tau hub ratios using the shuffled connectivity data. Mathematically, this exact test determines the percentage of shuffled beta values that fall below (i.e. for the association between younger age and a higher tau hub ratio) or above (i.e. for the association between a higher tau hub ratio and faster tau accumulation) the actual beta value, which can therefore be interpreted like a p-value. For the association between lower age and a higher tau hub ratio in symptomatic AD patients, we find a p-value of 0.016 for ADNI and of 0.019 for BioFINDER using the exact test. For the association between a higher tau hub ratio and faster tau accumulation in symptomatic AD, we found p-values of 0.010 in ADNI and 0.018 in BioFINDER. These additional analyses suggest that it is indeed the unique combination of the actual hub vector and the individual tau-PET data that drives our results, while any other linear combination of the connectivity data with tau-PET does not relate to younger age or faster tau accumulation in symptomatic AD. We have added these data to the results and statistics section of the manuscript (p.9, p.11, p.26).

(7) Hubness measures themselves are likely to have a strong age association. The results shown do not allow decoupling the fMRI from the tau to determine whether a similar trend could be revealed by hubness alone. This is a major concern. It would be helpful to perform this analysis in a subset of the ADNI data for which both tau PET and fMRI are available. That way the hubness effects and tau PET effects can be decoupled from each other and separately analyzed at an individual level. Such evidence would be more conclusive than what is provided.

Response: This is an interesting thought. However, we do not believe that hubness alone (i.e. without factoring in tau pathology) determines faster tau accumulation. Rather, we hypothesize that the richness of a hubs' connections should accelerate tau spreading once tau has reached a widely connected hub region (see Figure 4A), whereas tau in a non-hub region with less connections should ensue slower tau spreading due to the lower number of connections. Overall, we believe that a seed region harboring tau needs to be present before tau can actually spread, following the trans-neuronal tau spreading hypothesis.²⁰ Therefore, a pure "hubness" metric should not relate to faster tau accumulation in AD. To address the reviewers concern, we have downloaded and processed available resting-state fMRI data for ADNI subjects of the current cohort. A total of 77 resting-state fMRI datasets were available for Aβ+ subjects (preclinical AD, N=39, symptomatic AD=38), with uniform scanning protocols which were obtained within a year of the baseline tau-PET scan (3T EPI sequence, TR/TE/flip angle: 3000ms/30ms/90°, 200 volumes, 3.4mm isotropic voxel resolution). We preprocessed these resting-state fMRI data using the same pipeline as for HCP (see methods, p.22). Using the preprocessed fMRI data, we then determined subject-level hub maps using the same approach that is shown in Figure 1 for the HCP dataset. Specifically, we determined subject-level hub maps, i.e. we used subject-level functional connectivity matrices (i.e. Fisher z-transformed Pearson Moment correlations of preprocessed ROI timeseries), thresholded them at a density of 30%, transformed them to connectivity-based distance and computed ROI-wise weighted degree scores. To determine a subject-level summary score of overall "hub-ness", we averaged the weighted degree across all ROIs for each subject. Using linear regression, we did not find that the overall weighted degree score was associated with younger age (preclinical AD: $\beta = 0.096$, $p = 0.525$; symptomatic AD: $\beta = 0.162$, $p = 0.341$) or faster tau accumulation rates (preclinical AD: $\beta = -0.006$, $p = 0.973$; symptomatic AD: $\beta = -0.143$, $p = 0.415$). This supports the view that higher "hubness" alone is not associated with younger age or faster tau accumulation in symptomatic AD. Next, we used the individual connectivity data to determine subject-specific tau hub ratios, i.e. by combining subject-level tau-PET and subject-level resting-state fMRI. The correlation between tau hub ratios using individualized data and the tau hub ratios using HCP data was $r = 0.64$, suggesting a good correspondence between the tau hub ratio determined using individual connectivity vs. HCP connectivity data. Using this smaller dataset, we reran our main analyses on the association between younger age and a higher tau hub ratio, showing a significant association in symptomatic AD ($\beta = -0.307$, $p = 0.037$), but not in preclinical AD ($\beta = 0.11$, $p = 0.512$) in line with our main results using the HCP data in the larger tau-PET dataset. Similarly, we found an association between a higher tau hub ratio and faster tau accumulation in symptomatic AD ($\beta = 0.234$, $p = 0.025$) but not in preclinical AD ($\beta = 0.155$, $p = 0.331$). We have added these novel confirmatory analyses using subject-level fMRI data to the methods (p.21-22) and results section of the manuscript (p.9).

(8) The tau epicenters for analysis need to be named. The epicenters are currently described only as "10% of brain regions with highest baseline tau-PET positivity."

Response: We appreciate the reviewers' suggestion to have a more accurate description of the location of the epicenters. However, clearly labeling the epicenters is difficult, since the tau epicenters are defined for each patient individually, based on a given patients tau-PET pattern (i.e. 10% of brain regions with highest tau-PET defined for each patient individually). Therefore, there is not "one" group-level epicenter ROI but the epicenters show a different spatial pattern for each individual. Thus, we have performed a probability mapping of the tau epicenters in order to illustrate the spatial distribution of the tau epicenters in both cohorts. This probability mapping of the epicenters is shown in supplementary figure 1.

(9) The results in Fig 4 again are separate for preclinical vs. symptomatic. The global tau change for symptomatic cases is higher than that for asymptomatic preclinical cases. So I would not be surprised to see similar associations for all baseline tau PET measures, not just hubness-weighted ones. For this to be convincing, appropriate

references need to be reported. Also, this phrase needs to be justified: “tau epicenters fall in a hub region with many connections to other regions.” This analysis is not done for the specific epicenters or specific fMRI hubs. This is for all ROIs and is capturing a mixed effect of tau and hubness. So, I am not sure we are seeing effects specific to individual tau epicenters or fMRI hubs in these results.

Response: *To address the reviewers concern, we have repeated our analysis using shuffled hub maps in order to determine whether any other weighting scheme of tau-PET may also predict faster tau accumulation rates. Please see our response on comment 6 where we describe these analyses in detail. In addition, we agree that other tau-PET metrics (i.e. global tau-PET or temporal lobe tau-PET) should predict future tau accumulation, as shown previously by others.²¹ However, this study was specifically designed to assess whether the pattern of tau deposition (i.e. in hubs vs. non-hubs) held information on the rate of future tau accumulation and we do not claim that the tau hub ratio outperforms other tau-PET metrics or biomarkers in predicting future tau accumulation. Therefore, we did not perform a systematic comparison of the predictive accuracy of the tau hub ratio vs. other tau-PET metrics such as global or temporal lobe tau-PET for future tau accumulation. Please note also that our conclusion of a higher tau hub ratio being predictive of faster tau accumulation is further strengthened by our new sliding window analyses, showing that a higher tau hub ratio is associated with faster tau accumulation at a given level of baseline tau-PET. For details on these analyses, please refer to our response on comment 5 by reviewer 1 as well as supplementary figure 2.*

(10) While ADNI data can be accessed through their website, the terms for BioFINDER data sharing are very restrictive and would require contacting Dr. Hansson personally. To ensure reproducibility, it is recommended that scripts and data from both cohorts be made more easily accessible.

Response: *We have created a summary script with simulated data that allows to recapitulate the code for our main analyses. This analysis code has been uploaded as a supplementary file. For BioFINDER, we are, however, unable to facilitate the data sharing process, since this is subject to ethical and legal regulations at Lund University (see our data availability statement).*

In summary, the idea that hub regions can have a strong impact on tau spread even when the tau load in these regions is small is intuitive and meaningful. But the interpretations are not necessarily saying what the plots are showing. Phrases like “stronger shift of tau pathology towards globally connected hub regions” and “stronger bias of tau pathology towards more globally connected hubs” need to be removed from this manuscript. Results based on population-level hubness weights need to be compared with alternative weighting schemes for benchmarking. Finally, evidence from subjects with both fMRI and tau PET available would greatly increase the value of this study.

Response: *We would like to thank the reviewer for his comments and are confident that the additional analyses included in the revised version of the manuscript strengthen the view that hub regions in the brain may contribute to the spreading of tau pathology and facilitate earlier disease manifestation and faster progression of tau pathology.*

References:

- 1 Pawlik, D., Leuzy, A., Strandberg, O. & Smith, R. Compensating for choroid plexus based off-target signal in the hippocampus using (18)F-flortaucipir PET. **Neuroimage** 221, 2020.
- 2 Lace, G. *et al.* Hippocampal tau pathology is related to neuroanatomical connections: an ageing population-based study. **Brain** 132, 2009.
- 3 Berron, D. *et al.* Early stages of tau pathology and its associations with functional connectivity, atrophy and memory. **Brain** 144, 2021.
- 4 Lowe, V. J. *et al.* Widespread brain tau and its association with ageing, Braak stage and Alzheimer's dementia. **Brain** 141, 2018.
- 5 Mecca, A. P. *et al.* Association of entorhinal cortical tau deposition and hippocampal synaptic density in older individuals with normal cognition and early Alzheimer's disease. **Neurobiol Aging** 111, 2022.
- 6 Timmers, T. *et al.* Associations between quantitative [(18)F]flortaucipir tau PET and atrophy across the Alzheimer's disease spectrum. **Alzheimers Res Ther** 11, 2019.
- 7 La Joie, R. *et al.* Prospective longitudinal atrophy in Alzheimer's disease correlates with the intensity and topography of baseline tau-PET. **Sci Transl Med** 12, 2020.
- 8 Gordon, B. A. *et al.* Cross-sectional and longitudinal atrophy is preferentially associated with tau rather than amyloid beta positron emission tomography pathology. **Alzheimers Dement (Amst)** 10, 2018.
- 9 Sintini, I. *et al.* Longitudinal tau-PET uptake and atrophy in atypical Alzheimer's disease. **NeuroImage. Clinical** 23, 2019.
- 10 Lee, W. J. *et al.* Regional Abeta-tau interactions promote onset and acceleration of Alzheimer's disease tau spreading. **Neuron**, 2022.
- 11 Benson, G. *et al.* Functional connectivity in cognitive control networks mitigates the impact of white matter lesions in the elderly. **Alzheimers Res Ther** 10, 2018.
- 12 Franzmeier, N. *et al.* Left frontal cortex connectivity underlies cognitive reserve in prodromal Alzheimer disease. **Neurology** 88, 2017.
- 13 Franzmeier, N. *et al.* Left frontal hub connectivity delays cognitive impairment in autosomal-dominant and sporadic Alzheimer's disease. **Brain** 141, 2018.
- 14 Neitzel, J., Franzmeier, N., Rubinski, A., Ewers, M. & Alzheimer's Disease Neuroimaging, I. Left frontal connectivity attenuates the adverse effect of entorhinal tau pathology on memory. **Neurology** 93, 2019.
- 15 Cascone, A. D., Langella, S., Sklerov, M. & Dayan, E. Frontoparietal network resilience is associated with protection against cognitive decline in Parkinson's disease. **Commun Biol** 4, 2021.
- 16 Cole, M. W., Yarkoni, T., Repovs, G., Anticevic, A. & Braver, T. S. Global connectivity of prefrontal cortex predicts cognitive control and intelligence. **J Neurosci** 32, 2012.
- 17 Franzmeier, N. *et al.* Resting-state global functional connectivity as a biomarker of cognitive reserve in mild cognitive impairment. **Brain Imaging Behav** 11, 2017.
- 18 Franzmeier, N. *et al.* Patient-centered connectivity-based prediction of tau pathology spread in Alzheimer's disease. **Sci Adv** 6, 2020.
- 19 Franzmeier, N. *et al.* Tau deposition patterns are associated with functional connectivity in primary tauopathies. **Nat Commun** 13, 2022.
- 20 Wu, J. W. *et al.* Neuronal activity enhances tau propagation and tau pathology in vivo. **Nat Neurosci** 19, 2016.
- 21 Leuzy, A. *et al.* Biomarker-Based Prediction of Longitudinal Tau Positron Emission Tomography in Alzheimer Disease. **JAMA Neurol** 79, 2022.

REVIEWER COMMENTS

Reviewer #1 (Remarks to the Author):

The authors have done an impressive work including a set of additional analyses to answer, deeply and convincingly, to each of my comment. I am satisfied with the current version and congratulate the authors for their extensive work and what I consider as a very elegant contribution to the field.

Reviewer #2 (Remarks to the Author):

All my concerns have been addressed. The new terminology addresses the many ambiguities in the original submission. The inclusion of individual connectivity results is a strength. My only suggestion is to include a histogram of FreeSurfer ROIs with a high likelihood of being an epicenter in the supplement. This would be helpful for anyone who is trying to replicate this work as the regions can be identified by name. The surface heat map is not very helpful in this regard.

REVIEWERS' COMMENTS

Reviewer #1 (Remarks to the Author):

The authors have done an impressive work including a set of additional analyses to answer, deeply and convincingly, to each of my comment. I am satisfied with the current version and congratulate the authors for their extensive work and what I consider as a very elegant contribution to the field.

Response: We thank the reviewer for these encouraging remarks!

Reviewer #2 (Remarks to the Author):

All my concerns have been addressed. The new terminology addresses the many ambiguities in the original submission. The inclusion of individual connectivity results is a strength. My only suggestion is to include a histogram of FreeSurfer ROIs with a high likelihood of being an epicenter in the supplement. This would be helpful for anyone who is trying to replicate this work as the regions can be identified by name. The surface heat map is not very helpful in this regard.

Response: We would like to thank the reviewer for the constructive previous review and for these encouraging comments. Further, we appreciate the reviewers' comment to label the location of the epicenters. However, we would like to note that we did not use the Freesurfer-based Desikan-Killiany atlas to define epicenters but rather used the Schaefer atlas which is based on an fMRI-derived rather than surface-based anatomical brain parcellation. Therefore, there is no 1:1 translation between both atlases and multiple Schaefer ROIs may fall in the same Freesurfer ROI, since the Schaefer ROIs are typically smaller. Please note that the source data of Supplementary Figure 1 which has been uploaded with the manuscript includes the index of ROIs as encoded in the Schaefer 200 ROI atlas, as well as the epicenter probability values. The Schaefer atlas is freely available online (https://github.com/ThomasYeoLab/CBIG/tree/master/stable_projects/brain_parcellation/Schaefer2018_LocalGlobal) to the research community, hence other researchers can use the atlas together with our epicenter probability values from the source data file in order to assess the specific location of the epicenters. Thus, we believe that all necessary information is already available to replicate our work. To further address the reviewers concern, we have tried our best to map the Schaefer atlas to the Freesurfer atlas, based on atlas labels within the MNI-normalized Mindboggle Version of the Freesurfer atlas. Specifically, we have assigned Schaefer ROIs to Freesurfer ROIs based on the highest number of voxels shared between any given two ROIs of both atlases. We have added these labels to the source data file for Supplementary Figure 1 and now explicitly mention in the manuscript that the anatomical locations can be found there. We hope that this addition of anatomical labels addresses the reviewers concern.